# MAGNet: Motif-Agnostic Generation of Molecules from Shapes

## Abstract

Recent advances in machine learning for molecules exhibit great potential for facilitating drug discovery from *in silico* predictions. Most models for molecule generation rely on the decomposition of molecules into frequently occurring substructures (motifs), from which they generate novel compounds. While motif representations greatly aid in learning molecular distributions, such methods struggle to represent substructures beyond their known motif set. To alleviate this issue and increase flexibility across datasets, we propose MAGNet, a graph-based model that generates abstract shapes before allocating atom and bond types. To this end, we introduce a novel factorisation of the molecules' data distribution that accounts for the molecules' global context and facilitates learning adequate assignments of atoms and bonds onto shapes. Despite the added complexity of shape abstractions, MAGNet outperforms most other graph-based approaches on standard benchmarks. Importantly, we demonstrate that MAGNet's improved expressivity leads to molecules with more topologically distinct structures and, at the same time, diverse atom and bond assignments.

## 1 Introduction

The role of machine learning (ML) models in generating novel compounds has grown significantly, finding applications in fields like drug discovery, material science, and chemistry (Bian & Xie, 2021; Butler et al., 2018; Choudhary et al., 2022; Hetzel et al., 2022; Moret et al., 2023). These models offer a promising avenue for efficiently navigating the vast chemical space and generating unique molecules with specific properties (Hoffman et al., 2022; Zhou et al., 2019). A key ingredient contributing to the success of these models is their ability to encode molecules in a meaningful way, often employing graph neural networks (GNNs) to capture the structural intricacies (Gilmer et al., 2017; Mercado et al., 2021; Shi et al., 2021). Moreover, the inclusion of molecular *fragments*, known as *motifs*, significantly influences the generation process by enabling the model to explicitly encode complex structures such as cycles. This contrasts with the gradual formation of ring-like structures from individual atoms, forming chains until both ends unite (Yang et al., 2022; Zhu et al., 2022).

In the context of fragment-based models, researchers have adopted various techniques to construct fragment vocabularies, which can be categorised into chemically inspired and data-driven approaches. For example, both JT-VAE (Jin et al., 2018) and MoLeR (Maziarz et al., 2022) adhere to a heuristic strategy that dissects molecules into predefined structures, primarily consisting of ring systems and acyclic linkers. However, the diverse appearances of molecular structures result in various challenges concerning the vocabulary. While the large fragments in JT-VAE do not generalise well to larger datasets, the number of fragments becomes an issue for MoLeR. Such challenges are not unique to heuristic fragmentation methods but also extend to data-driven approaches like PS-VAE (Kong et al., 2022) and MiCaM (Geng et al., 2023). These approaches can set the number of chosen fragments but often resort to representing cyclic structures by combinations of chains. MiCaM, in particular, takes an approach that additionally incorporates attachment points for each fragment, resulting in a vocabulary that is "connection-aware", increasing its size by a significant margin. This leads to a situation where the included fragments, or motifs, often fall short to comprehensively represent the full spectrum of molecules present in the datasets (Sommer et al., 2023). Consequently, a generative model must refer to individual atoms in order to generate uncommon structures, a demanding task as the infrequent structures are often also more complex.

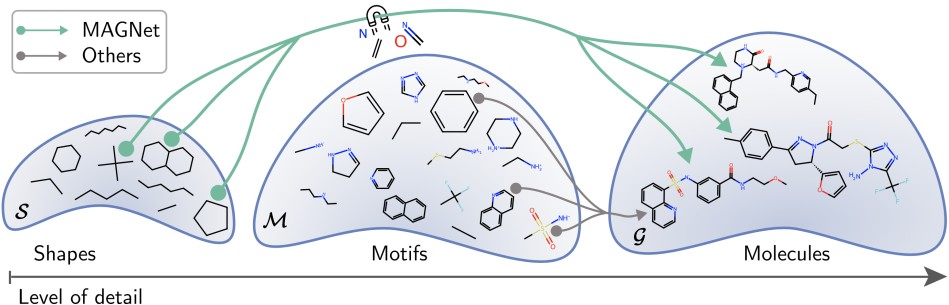

Figure 1: Current methods use motifs to compose new molecules. MAGNet advances this perspective and abstracts molecular subgraphs to untyped shapes. Being agnostic to atom and bond features in the beginning of the generation process, MAGNet generates molecules with great structural variety.

In this work, we address the limited structural diversity of existing fragment-based methods by abstracting motifs to their more general representation: **shapes**, see Fig. 1. By disentangling connectivity and features, i.e. atom and bond types, we reduce the combinatorial complexity needed to capture the entire distribution of molecules in a dataset, thus generating more diverse molecular structures. To enable the generation of molecules from such abstractions, we propose a novel factorisation of the data distribution, which splits a molecule into distinct components and detail how this facilitates constructing a concise vocabulary of shapes in § 2. Current molecule generation architectures cannot be trivially adapted to shapes, as they rely on motifs as building blocks. Additionally, most of the literature focuses on sequentially building up a molecule, conditioning on intermediate results at every step, see § 3. Such generation approaches, however, do not readily translate to shapes, as the atom and bond features of a particular shape should depend on the *entire* context of the molecule. Therefore, we propose a hierarchical generation approach, MAGNet, that first generates a molecule's abstract shape graph before defining its atom-level representation.

In summary, our contributions are:

- We address the issue of limited structural expressivity by abstracting motifs to shapes and using this abstraction to create a more flexible vocabulary.
- To match this abstraction, we propose MAGNet, an effective generation procedure that learns to generate molecules from abstract shape sets in a hierarchical fashion.
- Notably, our model is the first to freely featurise shapes, enabling it to sample a greater variety of atom and bond attributes than fragment-based approaches.

In our experiments, we evaluate (i) MAGNet's ability to sample diverse structures § 4.1, (ii) its generative performance according to established benchmarks § 4.2, (iii) the ability to generate adequate atom and bond features given the shape graph § 4.3, and (iv) the advantages of the proposed abstraction and hierarchical generation for downstream tasks such as conditional generation § 4.4.

## 2   MODELLING MOLECULES FROM SHAPES

We start by defining all mathematical objects and presenting the factorisation of the data distribution of molecular graphs, denoted as $\mathbb{P}(\mathcal{G})$. Following this, we will introduce our novel approach to identifying a concise set of shapes from data and present the corresponding MAGNet model.

**Factorising the data distribution** $\mathbb{P}(\mathcal{G})$   A molecular graph $\mathcal{G}$ is defined by its structure together with its node and edge features, describing atoms and bonds, respectively. In this work, we consider a factorisation of $\mathbb{P}(\mathcal{G})$ that decouples a molecular graph's topology from its features. For this, we build the molecular graph $\mathcal{G}$ as

$$\mathbb{P}(\mathcal{G}) = \mathbb{P}(\mathcal{G} \mid \mathcal{G}_{\mathcal{S}})\,\mathbb{P}(\mathcal{G}_{\mathcal{S}})\,,$$

where $\mathcal{G}$ refers to the full atom-level molecular graph and $\mathcal{G}_{\mathcal{S}}$ to its corresponding abstraction: the shape graph. $\mathcal{G}_{\mathcal{S}}$ represents a coarse view of a molecule's topology by specifying the shapes that

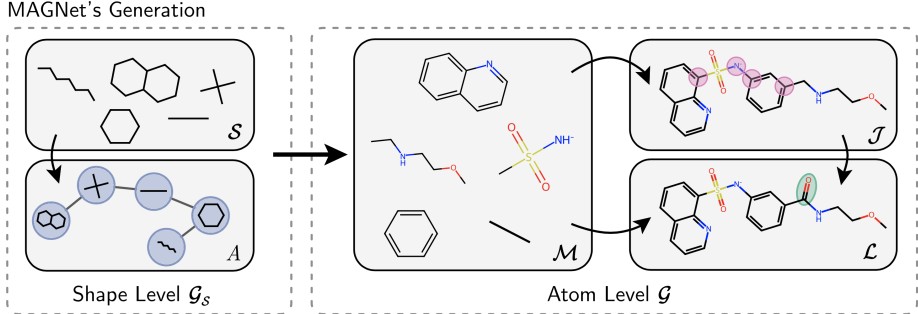

Figure 2: On the shape-level, MAGNet predicts the shape multiset $\mathcal{S}$ and its connectivity $A$. Progressing to the atom-level, $\mathcal{G}_{\mathcal{S}}$ informs the generation of shape representations $\mathcal{M}$. To fully define the molecular graph $\mathcal{G}$, the join node positions $\mathcal{J}$ and the leaf nodes $\mathcal{L}$ are predicted.

make up the molecule as well as their connectivity. Thus, $\mathbb{P}(\mathcal{G}_{\mathcal{S}})$ can be factorised further into the multiset of shapes $\mathcal{S}$ and their connectivity $A \in \mathcal{A}^{|\mathcal{S}| \times |\mathcal{S}|}$, where $\mathcal{A}$ defines the possible values of $A$.

Moving forward to the atom level, a node $S$ in the shape graph $\mathcal{G}_{\mathcal{S}}$ can be expanded into its binary adjacency matrix of its $s = |S|$ nodes, i.e. $S \in \{0,1\}^{s \times s}$ and is equipped with node and edge features, corresponding to atom and bond types, respectively. We consider this feature-equipped representation of $S$ to be a typed subgraph $M$, or motif, of the input graph $\mathcal{G}$, and denote the associated multiset within the molecule by $\mathcal{M}$.

The connectivity between two shapes, signified by $A_{ij} \neq 0$, indicates that, at the atom level, the two shapes share a common atom $j$. To determine $j$, we define the set of join nodes $\mathcal{J}$, which can be understood as the set of join nodes $j$ that are contained in two motifs: $\mathcal{J} = \{j \mid j \in M_k, \ j \in M_l, \ A_{kl} \neq 0, \ S_k, S_l \in \mathcal{S}\}$. Note that we can effectively provide information about the atom type of the join node $j$ by considering $\mathcal{A} = \{0, C, N, \dots\}$ to include atom types, see Appendix C.4. Lastly, to allow for a concise set of structurally distinct shapes, not all atoms are described by the shape graph. Specifically, we define the set of leaf nodes $\mathcal{L}$ to include nodes $l$ with degree $d_l = 1$ and neighbours $k \in \mathcal{N}_l$ with $d_k = 3$.

In conclusion, $\mathbb{P}(\mathcal{G})$ is factorised into the motifs $\mathcal{M}$, the join nodes $\mathcal{J}$, and the leaf nodes $\mathcal{L}$:

$$\mathbb{P}(\mathcal{G}) = \mathbb{P}(\mathcal{L}, \mathcal{J}, \mathcal{M})$$
$$= \mathbb{P}(\mathcal{L} \mid \mathcal{M}, \mathcal{J})\mathbb{P}(\mathcal{J} \mid \mathcal{M}, A)\,\mathbb{P}(\mathcal{M} \mid \mathcal{G}_{\mathcal{S}})\,\mathbb{P}(\mathcal{G}_{\mathcal{S}})\,,$$

where we use $\mathcal{G}_{\mathcal{S}}$ to factorise the distribution. Note that $\mathcal{J}$ is *conditionally independent* of $\mathcal{S}$ given the motifs $\mathcal{M}$. The same applies to $\mathcal{L}$ being conditionally independent of $A$ given $\mathcal{J}$.

## 2.1 IDENTIFYING A CONCISE SET OF SHAPES FROM DATA

The specific set of shapes on which a model is trained has to be determined from data. Given a dataset of molecules, our fragmentation scheme aims to represent a molecule through clear structural elements, for which we provide examples in Appendix A. For this, we start by removing all leaf nodes $\mathcal{L}$ across the graph, following the approach outlined in previous works (Jin et al., 2020; Maziarz et al., 2022). This step helps to divide the molecule $\mathcal{G}$ into cyclic and acyclic parts. Importantly, instead of modelling the connection between two fragments $M_i$ and $M_j$ with a connecting bond, we represent it by a shared atom, referred to as the join node $\nu \in \mathcal{J}$.

Moreover, we further decompose the resulting acyclic fragments to reduce the number of required shapes as much as possible. To this end, we introduce "junctions" defined by a node and its neighbours, with degree three or four present in an acyclic structure. On the ZINC dataset (Irwin et al., 2020), consisting of 249,456 compounds, our fragmentation results in 7371 typed subgraphs. When compared to the "Breaking Bridge Bonds" decomposition (Jin et al., 2020; Maziarz et al., 2022), our approach reduces complexity by collapsing acyclic structures into distinct shapes, cutting the vocabulary size in half. Additionally, in contrast to data-driven methods like those outlined in Kong et al. (2022) and Geng et al. (2023), our decomposition method maintains structural integrity through a top-down approach.

These fragments are typed, and MAGNet abstracts them further into shapes by removing atom and bond types, resulting in 347 distinct shapes. In the best case, this process consolidates up to 800 motifs into a single shape. As a suitable generative model has to to map a single shape to its various representations, this fragmentation further enables us to model smoother transitions between shape representations. This is in contrast to fragment-based methods, which have to select different tokens from a large vocabulary when two motifs differ in, e.g. just bond type, see Fig. 6 b in Appendix A.

## 2.2 MAGNET'S GENERATION PROCESS

MAGNet is designed to represent the hierarchy of the factorisation into shape- and atom-level from § 2, cf. Fig. 2. The model is trained as a VAE model (Kingma et al., 2015), where the latent vectors $z$ are trained to encode meaningful semantics about the input data. That is, given a vector $z$ from the latent space, MAGNet's generation process first works on the shape level to predict $\mathcal{G}_\mathcal{S}$, defined by the multiset $\mathcal{S}$ and its connectivity $A$, before going to the atom level defined by the motifs $\mathcal{M}$, join nodes $\mathcal{J}$, and leaf nodes $\mathcal{L}$. Additional details on the implementation can be found in Appendix B.2.

**Shape-Level**    On the shape-level, MAGNet first generates the **shape multiset** $\mathcal{S}$—the same shape can occur multiple times in one molecule—from the latent representation $z$. More specifically, we learn $\mathbb{P}(\mathcal{S} \mid z)$ by conditioning the generation on the latent code $z$ and generate one shape at a time, conditioning also on the intermediate representation of the shapes.

Given the shape multiset $\mathcal{S}$, MAGNet infers the **shape connectivity** $A$ between shapes $S_i, S_j \in \mathcal{S}$. Formally, we learn $\mathbb{P}(A \mid \mathcal{S}, z) = \prod_{i,j=1}^{n} \mathbb{P}(A_{ij} = t \mid S, z)$ where $t \in \{0, C, N, \dots\}$ not only encodes the existence (or absence) of a shape connection but also its atom type. We consider a typed version of $A$ to provide a meaningful condition for generating the shape representations $\mathcal{M}$. We assume the individual connections, $A_{ij}$ and $A_{lk}$, to be independent given the shape multiset $\mathcal{S}$ and the latent representation $z$. The loss on the shape level is computed by $L_{\mathcal{G}_\mathcal{S}} = L_\mathcal{S} + L_A$, where $L_\mathcal{S}$ and $L_A$ refer to the categorical losses of the shape set $\mathcal{S}$ and connectivity $A$, respectively.

**Atom-Level**    Leveraging $\mathcal{S}$ and $A$, which together define a molecule's shape-level representation, MAGNet transitions to the atom-level by discerning appropriate node and edge attributes for each shape. By predicting the **atom and bond types** the motifs $\mathcal{M}$ are defined. To model the shape representation $\mathbb{P}(M_i \mid \mathcal{S}, A, z)$ of shape $S_i$, we use the encoded shape graph $\mathcal{G}_\mathcal{S}$ together with the latent code $z$ and learnable embeddings for each shape $S_i$ to predict the respective atom types $M_i^a$.

Subsequently, the resulting atom embeddings are leveraged to determine the corresponding **bond types** $M^b$ between connected nodes. Note that conditioning on $A$ ensures that $M_k$ includes all nodes required for connectivity also on the atom-level, i.e. the atom allocations for $M$ have to respect all join node types defined by the shape connectivity $A$: $A_{kl} \in \bigcup_j M_k^a \cap M_l^a$, where $a$ signifies the exclusive consideration of atoms.

To establish connectivity on the atom level, MAGNet proceeds to identify the **join nodes** $\mathcal{J}$. The join nodes' types are already determined by the connectivity $A$. MAGNet accomplishes this by predicting the specific node positions $p_a$ that need to be combined, collectively forming the set of join nodes $\mathcal{J}$. The likelihood of merging nodes $i$ and $j$ in shape representations $M_k$ and $M_l$ is represented by the merge probability $J_{ij}^{(k,l)} = \mathbb{P}(p_i \equiv p_j \mid \mathcal{M}, A, z)$, which constitutes the join matrix $J^{(k,l)} \in [0,1]^{V_{S_k} \times V_{S_l}}$.

Finally, predicting the **leaf nodes** $\mathcal{L}$ involves determining the correct atom type for each leaf node and attaching it to the molecule's atom representation, denoted as $\mathcal{C}$ for core molecule. To learn $\mathbb{P}(\mathcal{L}_S \mid \mathcal{C}, z)$, we assess each node position within the shape representation $M_S$. Optimising this likelihood is done in a similar fashion to the approach employed in modelling the shape representation $P(M_i \mid S, A, z)$ for shape $S_i$, only that the shape graph is replaced by the atom graph $\mathcal{C}$. The atom-level loss is defined by $L_\mathcal{G} = L_\mathcal{M} + L_\mathcal{J} + L_\mathcal{L}$, where $L_\mathcal{M}$, $L_\mathcal{J}$, and $L_\mathcal{L}$ describe categorical losses for shape features $\mathcal{M}$, including both atom and bond types, join nodes $\mathcal{J}$, and leaf nodes $\mathcal{L}$, respectively.

## 2.3 THE MAGNET ENCODER

MAGNet's encoder aims to learn the approximate posterior $\mathbb{Q}(z \mid \mathcal{G})$. At its core, the encoder leverages a graph transformer Shi et al. (2021) for generating node embeddings of the molecular

graph. Since MAGNet generates molecules in a coarse to fine-grained fashion, it is beneficial to encode information about the decomposition of the molecules. To achieve this, we use an additional GNN to capture the coarse connectivity within the shape graph. The embeddings of $\mathcal{G}$ and $\mathcal{G}_\mathcal{S}$ are computed by aggregating over the individual atom and shape nodes, respectively. In addition to these aggregations, we separately aggregate join and leaf nodes to represent the other essential components of the graph. The representation of the individual components—molecular graph, shapes, join nodes, and leaf nodes—are concatenated and mapped to the latent space by an MLP, constituting the graph embedding $z_\mathcal{G}$. More details about the chosen node features as well as technical specifications of the encoder can be found in Appendix B.1.

Taken together, we optimise MAGNet according to the VAE setting, maximising the ELBO:

$$
\begin{aligned}
L &= \mathbb{E}_{z \sim \mathbb{Q}}\big[\mathbb{P}\big(\mathcal{G} \mid z\big)\big] + \beta D_{\text{KL}}\big(\mathbb{Q}(z \mid \mathcal{G}) \mid P\big) \qquad \text{with} \quad P \sim \mathcal{N}(0, \mathbb{1}) \\
&= L_{\mathcal{G}_\mathcal{S}} + L_\mathcal{G} + \beta D_{\text{KL}} \,,
\end{aligned}
$$

where the KL-divergence $D_{\text{KL}}$ serves to regularise the posterior $\mathbb{Q}(z \mid \mathcal{G})$ towards similarity with the Normal prior $P$ in the latent space, weighted by $\beta$. Training MAGNet, we have observed that this regularisation alone is inadequate for achieving a smoothly structured latent space as the latent space suffers from over-pruning behaviour (Yeung et al., 2017), see our analysis in Appendix B.3. To remedy this, we apply a normalising flow post-hoc to the latent space, aligning it more effectively with the prior. To this end, we rely on Conditional Flow Matching (Lipman et al., 2023) and, more specifically, use the version based on minibatch optimal transport as presented by Tong et al. (2023). We specify MAGNet's hyperparameter configuration in Appendix B.4.

## 3  RELATED WORK

**Molecule generation**   Existing generative models can be divided into three categories (Du et al., 2022; Yang et al., 2022; Zhu et al., 2022): (1) string-based models, relying on string representations like SMILES or SELFIES (Adilov, 2021; Fang et al., 2023; Flam-Shepherd et al., 2022; Grisoni, 2023; Gómez-Bombarelli et al., 2018; Segler et al., 2018), which do not leverage structural information, (2) graph-based models, which model the molecular graphs, and (3) geometry-based models, which represent molecules by atomic point clouds (Garcia Satorras et al., 2021; Gebauer et al., 2019; 2022; Hoogeboom et al., 2022; Huang et al., 2023a;b; Luo et al., 2021a; Luo & Ji, 2022; Qiang et al., 2023; Ragoza et al., 2020; Vignac et al., 2023; Xu et al., 2023). Besides these representation techniques, molecules can also be represented by molecular descriptors. However, these rely on irreversible hashing which makes them unsuited for generation tasks (Du et al., 2022; Jiang et al., 2021).
Graph-based approaches involve models that represent molecular graphs (i) primarily at the atom level or (ii) predominantly through motifs. Zhu et al. (2022) categorise the generation process further into *sequential* methods (Ahn et al., 2021; Assouel et al., 2018; Bengio et al., 2021; Jin et al., 2020; Kajino, 2019; Khemchandani et al., 2020; Li et al., 2018; Lim et al., 2020; Liu et al., 2018; Luo et al., 2021b; Mercado et al., 2021; Popova et al., 2019; Shi et al., 2020; Shirzad et al., 2022; Yang et al., 2021; You et al., 2019), building molecules per fragment while conditioning on a partial molecule, and *one-shot* (OS) approaches (Bresson & Laurent, 2019; De Cao & Kipf, 2018; Flam-Shepherd et al., 2020; Kong et al., 2022; Liu et al., 2021; Ma et al., 2018; Samanta et al., 2019; Simonovsky & Komodakis, 2018; Zang & Wang, 2020) that create each aspect of the molecular graph in a single step. Note that diffusion-based models iteratively refine the entire graph, making them difficult to categorise as sequential or one-shot. While these models are predominantly used in the 3D context, Vignac et al. (2022) propose a discrete diffusion process that falls into category (2).

**Fragmentation and shape representation**   Various techniques are available for constructing fragment vocabularies, with a distinction between chemically-inspired and data-driven approaches. For example, both HierVAE (Jin et al., 2020) and MoLeR (Maziarz et al., 2022) adopt a heuristic strategy known as "breaking bridge bonds" to decompose molecules into rings and remainder fragments, emphasising chemically valid substructures. In a similar vein, JT-VAE (Jin et al., 2018) employs fragmentation guided by the construction of junction trees. In contrast, PS-VAE (Kong et al., 2022) and MiCaM (Geng et al., 2023) take a data-driven bottom-up approach, creating fragments by merging smaller components, starting from single atoms. MiCaM even integrates attachment points, resulting in a larger, "connection-aware" vocabulary.

MAGNet is a graph-based model that employs a unique approach by generating each hierarchy level in a single step, similar to existing OS approaches. It positions itself between the traditional categories of single-atom and fragment-based models by utilising shapes as building blocks and subsequently generating appropriate atom and bond attributes. Our approach to defining shapes is based on the topological properties of a molecular graph. This differentiates our work from others (Adams & Coley, 2022; Chen et al., 2023; Long et al., 2022) who refer to shapes within a 3D context and consider surface areas as generation targets. Our fragmentation approach is designed to achieve concise topological representations, enabling to generate diverse structures from a limited vocabulary.

## 4 EXPERIMENTS

We evaluate MAGNet's performance across several dimensions of the generation process. In § 4.1, we investigate the reconstruction and sampling of shapes $\mathcal{S}$, as the fundamental component of MAGNet's factorisation, and assess how faithfully our model represents the diverse structural characteristics found in molecules. In § 4.2, we continue to evaluate the generative performance using established benchmarks. Next, see § 4.3, we analyse MAGNet's ability to determine suitable atom and bond allocations $\mathcal{M}$, highlighting its distinctive approach compared to baseline models. In § 4.4, we demonstrate how the presented factorisation and shape vocabulary facilitate MAGNet to generalise effectively across different datasets in a zero-shot manner. Finally, we explore the possibilities enabled by our factorisation, such as conditioning on various levels of the generation process.

### 4.1 USING SHAPES TO REPRESENT STRUCTURAL VARIETY OF MOLECULES

**Reconstructing complex structures**   Our first experiment provides qualitative insights into how accurately shapes are decoded, $\mathbb{P}(\mathcal{S} \mid z_{\mathcal{G}})$. For this experiment, we employ two baselines: PS-VAE (Kong et al., 2022) and MoLeR (Maziarz et al., 2022), the models with the best performance within their respective category on the GuacaMol Benchmark, see § 4.2. We assess the decoder's performance in reconstructing molecules from the test set, which includes uncommon shapes like large rings or complex junctions. Our observations reveal that the baseline models have difficulty in constructing complex shapes, as illustrated in Figure 3a. This limitation is likely attributed to the absence of such shapes in their top-$k$ vocabularies. Consequently, these models face the challenge of constructing shapes such as large rings from individual atoms. In contrast, our proposed model, MAGNet, operates with a moderately-sized shape vocabulary that includes complex shapes, enabling it to generate molecules that closely adhere to the latent code and the corresponding ground truth molecules. We quantify this result through the displacement of latent codes in Appendix C.1.

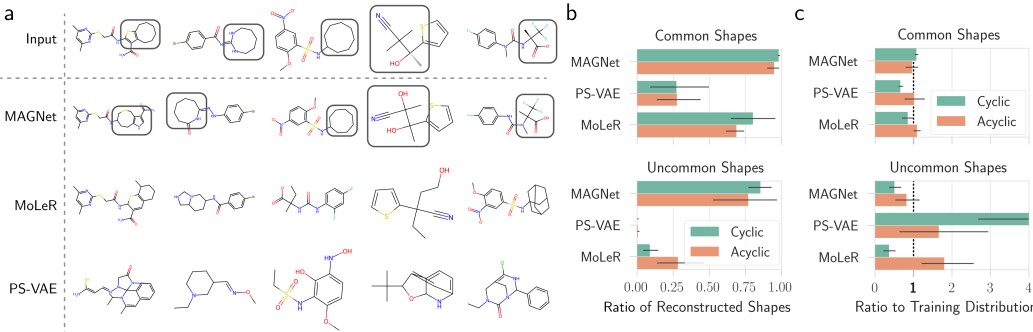

Figure 3: **(a)** Reconstruction of molecules that include large cycles or complex junctions. Relying on individual atoms to build these structures is not sufficient. Only MAGNet is able to reliably decode its latent code $z$. **(b)** Percentage of reconstructed shapes. MAGNet substantially improves in reconstructing both common and—more importantly—uncommon shapes. **(c)** Comparison of sampled shapes to shape occurrences in the training distribution. A ratio of 1 is optimal.

**MAGNet reliably decodes shapes**  Building on our analysis of large cycles and uncommon junctions, we extend our investigation to assess how effectively different models can reconstruct the shape set $\mathcal{S}$ in a general context. Given our focus on $\mathbb{P}(\mathcal{S} \mid z)$, we can disregard the shape connectivity $A$ and representations $\mathcal{M}$. As illustrated in Fig. 3b, our findings demonstrate that MAGNet consistently outperforms both MoLeR and PS-VAE. Although it is to some extent expected from the design of MAGNet to have enhanced expressivity of shapes, our results support the hypothesis that the other methods fail to learn the concept of a shape, relying primarily on the information encoded in their vocabulary. Note that the ability to faithfully represent shapes simultaneously requires MAGNet to freely learn their features, which we analyse in § 4.3.

**MAGNet matches the distribution of shapes more accurately**  To further check that uncommon shapes are also sampled, we analyse the shape set of generated molecules. If the other models are able to represent scaffolds that are not included in their vocabulary, they should be able to reflect the reference distribution of shapes. Fig. 3c shows that this is *not* the case in practice. For this evaluation, we decompose sampled molecules into their shapes. We then measure the models' over- and undersampling behaviour based on the ratio $r_{S_i} = \frac{c_s(S_i)}{\sum_k c_s(S_k)} \times \frac{\sum_k c_t(S_k)}{c_t(S_i)}$, where $c_t$ and $c_s$ refer to the count function applied to the training and sampled sets, respectively. On common shapes, i.e. those that occur in more than 10% of the molecules, all evaluated models are able to match the ratio of the ZINC distribution. For uncommon shapes, however, both MoLeR and PS-VAE fail: while PS-VAE heavily oversamples both ring-like structures and chain-like structures, MoLeR oversamples chain-like structures and undersamples ring-like structures. MAGNet matches the reference distribution best across categories and we conclude that the proposed abstraction to shapes is also beneficial for generation.

## 4.2 GENERATIVE PERFORMANCE EVALUATED ON COMMON BENCHMARKS

Employing two standard benchmarks for de-novo molecule generation, we establish MAGNet's competitive generative performance. The GuacaMol benchmark asseses the ability of a generative model to sample in accordance with the distribution of a molecular dataset (Brown et al., 2019). Next to evaluating the uniqueness and novelty of sampled molecules, the benchmark also computes distributional distances to the reference, i.e. the KL-divergence and Fréchet distance (FCD). We use the MOSES benchmark (Polykovskiy et al., 2020) to report measures for the internal diversity (IntDiv) of generated molecules as well as chemical properties such as synthetic accessability (SA), the octanol-water partition coefficient (logP), and the viability for drugs (QED). Baselines for these benchmarks additionally include JTVAE (Jin et al., 2018), HierVAE (Jin et al., 2020), and MiCaM (Geng et al., 2023) as sequential, fragment-based methods. For those model with a variable vocabulary, we set the size to 350. We also include GraphAF (Shi et al., 2020) as a purely atom-based model. While the focus of this work lies on graph-based molecule generation, we furthermore add the SMILES-based baseline of the GuacaMol benchmark SMILES-LSTM (SM-LSTM) (Segler et al., 2018) as well as the SMILES-based VAE CharVAE (Gómez-Bombarelli et al., 2018). Importantly, SM-LSTM does not have a latent space and can thus not perform targeted decoding. For all baselines, we use the hyperparameters specified in their respective works.

Table 1: GuacaMol and MOSES Benchmark. We report mean and standard deviation using 5 random seeds and highlight the best overall graph-based method as well as **the best within each category.**

| | | GuacaMol | | MOSES | | |
|---|---|---|---|---|---|---|
| | | FCD ($\uparrow$) | KL ($\uparrow$) | IntDiv ($\uparrow$) | logP ($\downarrow$) | SA ($\downarrow$) | QED ($\downarrow$) |
| SM. | CharVAE | $0.17 \pm 0.08$ | $0.78 \pm 0.04$ | $\mathbf{0.88} \pm \mathbf{0.01}$ | $0.87 \pm 0.14$ | $0.48 \pm 0.13$ | $0.06 \pm 0.03$ |
| | SM.-LSTM | $\mathbf{0.93} \pm \mathbf{0.00}$ | $\mathbf{1.00} \pm \mathbf{0.00}$ | $0.87 \pm 0.00$ | $\mathbf{0.12} \pm \mathbf{0.01}$ | $\mathbf{0.04} \pm \mathbf{0.02}$ | $\mathbf{0.00} \pm \mathbf{0.00}$ |
| Sequential | GraphAF | $0.05 \pm 0.00$ | $0.67 \pm 0.01$ | $\mathbf{0.93} \pm \mathbf{0.00}$ | $0.41 \pm 0.02$ | $0.88 \pm 0.10$ | $0.22 \pm 0.01$ |
| | HierVAE | $0.53 \pm 0.14$ | $0.92 \pm 0.01$ | $0.87 \pm 0.01$ | $0.36 \pm 0.17$ | $0.20 \pm 0.14$ | $0.03 \pm 0.00$ |
| | MiCaM | $0.63 \pm 0.02$ | $0.94 \pm 0.00$ | $0.87 \pm 0.00$ | $0.20 \pm 0.05$ | $0.51 \pm 0.03$ | $0.08 \pm 0.00$ |
| | JTVAE | $0.75 \pm 0.00$ | $0.94 \pm 0.00$ | $0.86 \pm 0.00$ | $0.28 \pm 0.03$ | $0.34 \pm 0.01$ | $\underline{\mathbf{0.01}} \pm \underline{\mathbf{0.00}}$ |
| | MoLeR | $\mathbf{0.80} \pm \mathbf{0.01}$ | $\mathbf{0.98} \pm \mathbf{0.00}$ | $0.87 \pm 0.00$ | $\underline{\mathbf{0.13}} \pm \underline{\mathbf{0.02}}$ | $\mathbf{0.06} \pm \mathbf{0.01}$ | $\underline{\mathbf{0.01}} \pm \underline{\mathbf{0.01}}$ |
| OS | PSVAE | $0.28 \pm 0.01$ | $0.83 \pm 0.00$ | $\mathbf{0.89} \pm \mathbf{0.00}$ | $0.34 \pm 0.02$ | $1.18 \pm 0.05$ | $0.05 \pm 0.00$ |
| | MAGNet | $\mathbf{0.76} \pm \mathbf{0.00}$ | $\mathbf{0.95} \pm \mathbf{0.00}$ | $0.88 \pm 0.00$ | $\mathbf{0.22} \pm \mathbf{0.01}$ | $\mathbf{0.12} \pm \mathbf{0.01}$ | $\underline{\mathbf{0.01}} \pm \underline{\mathbf{0.00}}$ |

**MAGNet is the best OS model on standard benchmarks**   The benchmark is conducted on $10^4$ latent codes sampled from the prior distribution, $z \sim P$, and decoded into valid molecules. Our results for both benchmarks on the ZINC dataset are depicted in Table 1, where we classify the methods into their generative approaches as described in § 3. We do not report Novelty and Uniqueness, as almost all evaluated models achieve 100% on these metrics. Solely GraphAF and HierVAE achieve 91% and 96% Uniqueness and Novelty, respectively. For baselines like SM-LSTM and CharVAE, which are not able to achieve 100% Validity, we sample until we obtain $10^4$ valid molecules. While MoLeR sets the state of the art on both FCD and KL, MAGNet overall performs competitively, outperforming all other graph-based baselines. This supports the proposed factorisation in § 2 while also challenging the common perception that methods for molecule generation must rely on motif vocabularies to obtain good generative performance.

**The FCD metric is insufficient for evaluating structural diversity**   Despite the FCD being an important metric for molecular distribution learning, we find that it fails to provide insights about the structural diversity of the generated molecules. Evaluating the benchmark on a subset of $10^4$ molecules from the training data, which was filtered to include only the 10 most common shapes, results in an FCD score of 0.89. This observation offers an explanation for why models like MoLeR can achieve state-of-the-art FCD scores, despite not accurately capturing the distribution of uncommon shapes, as demonstrated in § 4.1. This underscores that our evaluation of the structural diversity of molecule is orthogonal to these benchmarks, providing valuable insights into the tails of the molecular distribution.

### 4.3   GENERATION OF SHAPE REPRESENTATIONS $\mathcal{M}$

Having established MAGNet's ability to utilise its shape vocabulary to reliably decode a molecule's structure and sample diversely, we further evaluate MAGNet's atom and bond allocation to shapes.

**MAGNet's shape representations are superior to fixed fragments**   The larger a given shape, the more the combinatorial aspect starts to dominate: with a size-limited vocabulary, it is challenging to reflect the diversity of a shape's realisations during decoding. This is shown in Fig. 4a, which provides a qualitative view on shape representations. We extract shape representations of a given shape from the molecules sampled in Table 1 and plot the two principal components of their fingerprints. Only for this shape, there are 791 representations in ZINC. Both PS-VAE as well as MoLeR are not able to cover the distribution fully, even though the shape appears commonly in the dataset. MAGNet, by contrast, covers all parts of the distribution, even outliers. Fig. 4b shows the MMD quantification of the results in Fig. 4a, confirming that MAGNet is able to best cover the entire distribution of shape representations. Being able to *reliably* decode a large variety of molecular scaffolds is especially important for downstream tasks such a molecule optimisation.

**Allocation of atom and bonds to shapes**   Extending the sampling analysis from Fig. 4b, we quantify the process of turning an abstract shape into a chemically valid substructure in Fig. 4c.

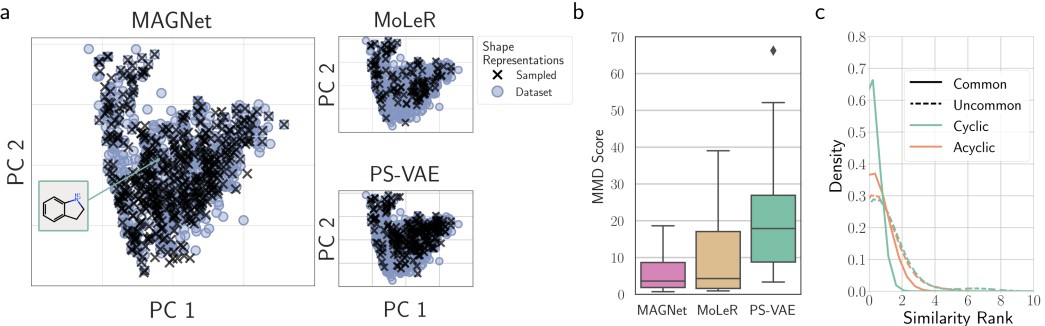

Figure 4: **(a)** Example of generated fragments by MAGNet and baseline methods. **(b)** MMD computation to quantify similarity between generated and ground truth shape representations. **(c)** Rank comparison between predicted fragments and their original counterparts.

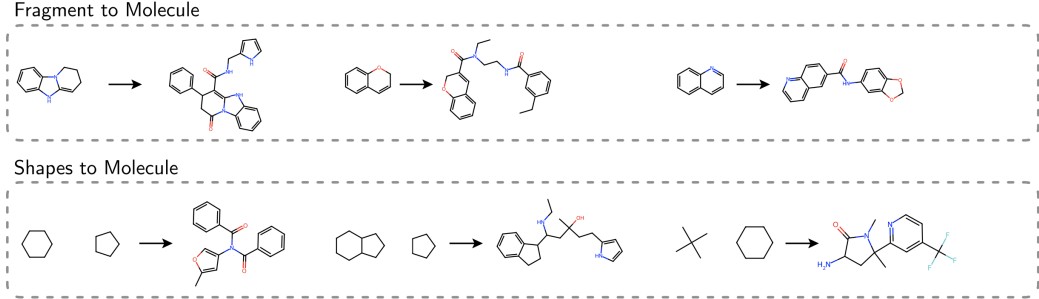

Figure 5: Examples of conditional molecule generation with MAGNet. The generation is conditioned on (*top*) a complete fragment, including atoms and edges, and (*bottom*) two distinct shapes.

For each shape in the ZINC dataset, we compute the similarities between the set of all predicted and ground truth allocations. Given a ground truth assignment and a successful shape decoding, we measure how the decoded allocation ranks compare to known allocations. In the majority of cases, MAGNet achieves rank $0$ or $1$ in the shape allocation, with uncommon rings being the most challenging to decode.

## 4.4 APPLICATION ACROSS DATASETS AND CONDITIONAL SAMPLING

Having analysed the generative performance of the MAGNet model and the benefit of the proposed shape fragmention, we continue to investigate how well the shape abstraction derived from the ZINC data translates to other datasets. After this, we showcase how one can use MAGNet's whole context generation for the generation of linkers and scaffold constrained generation.

**Shape abstractions translate well across datasets**   To examine the flexibility of our shape fragmentation, we evaluate its transferability to unseen datasets with distinct molecular distributions through zero-shot generalisation. We use the datasets QM9 (Wu et al., 2018), GuacaMol (Brown et al., 2019), CheMBL (Mendez et al., 2019), and L1000 (Subramanian et al., 2017). Note that we do not finetune any model on the unseen datasets and only use the vocabulary extracted from ZINC. MAGNet is able to achieve the highest similarity scores across all datasets, improving over the strongest baseline by up to 20%, see Appendix C.3 for more details. This underscores the flexibility of the fragmentation and MAGNet's expressive power across the space of drug-like molecules.

**MAGNet efficiently generates molecules conditioned on shapes and scaffolds**   In the context of potential downstream applications, we investigate novel scaffold conditioning methods made possible by MAGNet's factorisation. Besides the latent space interpolation in Appendix C.2, Fig. 5 illustrates that MAGNet is capable to condition not only on a single scaffold but also on multiple scaffolds, even when they are not directly connected within the resulting molecule. This poses a significant challenge for models like MoLeR, which rely on extending connected subgraphs for scaffold conditioning. Moreover, MAGNet enables conditioning on multiple levels and can generate molecules conditioned on a fragment as well as solely based on shapes, cf. Fig. 5 *(top)* and *(bottom)*, respectively. Conditioning only on shapes enables the free allocation of atoms and bonds—a form of conditioning that was previously not possible.

## 5 CONCLUSION

We present MAGNet, a generative model for molecules that relies on a novel factorisation to disentangle structure from features, thus leading to a general abstraction in the space of molecules. MAGNet exhibits stronger performance at representing structures than existing models while also showing favourable results in generative tasks. While we argue that a global context like the one adopted in MAGNet is important for shape representations, modifications thereof can also be promising for sequential models. Finally, our proposed abstraction to shapes lends itself to *general* applications in graph generative models beyond the molecular world.

**Reproducibility Statement** To ensure reproducibility we make available the code for MAGNet. Additional documentation is provided in Appendix B.4, including all hyperparameters and training specifications necessary to reproduce the results discussed in this work. Both training and inference work on readily-available hardware, and detailed computational requirements are outlined in Appendix B.4. All data used for the experimental evaluation is publicly available.

**Ethics Statement** We commit to full transparency by making our research code publicly available. Together with openly accessible datasets, this facilitates widespread utilisation of MAGNet. However, this comes with potential risks of misuse inherent to the field of drug discovery. The capacity to generate molecules can go beyond benevolent drug discovery and can inadvertently lead to the creation of hazardous compounds or substances with unforeseen consequences. These risks emphasise the necessity for responsible use and oversight in the application of our methodology. However, our work also holds the potential to advance drug discovery efforts, potentially aiding in the identification of new pharmaceutical compounds.

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

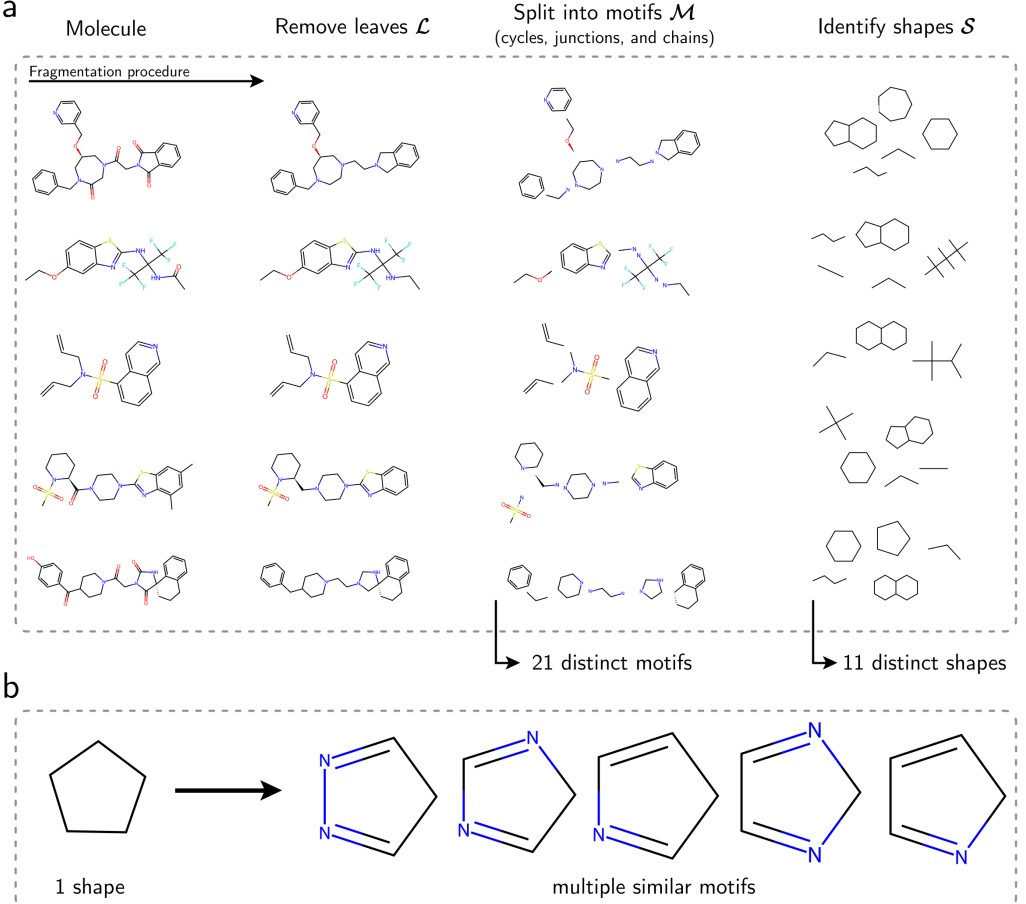

Figure 6: **(a)** Examples of the fragmentation procedure. Starting with the entire molecules, the fragmentation first removes leaf nodes $\mathcal{L}$, continues to split into motifs $\mathcal{M}$, and then identifies distinct shapes $\mathcal{S}$. **(b)** Example for a single shape that has multiple similar representations in terms of atom and bond types, illustrating how our shape abstraction can reduce combinatorial explosion and enable smooth learning.

## A  ABSTRACTION TO SHAPES AND DETAILS ON PROPOSED FRAGMENTATION

Fig. 6 a shows examples of the proposed fragmentation and abstraction to shapes. First, we identify leaf nodes $\mathcal{L}$ and then divide the core molecule $\mathcal{C}$ into structurally distinct fragments $\mathcal{M}$ that can be categorised into rings, chains, and junctions. Note that adjacent shapes share a join node $v \in \mathcal{J}$ instead of being connected through a bond. This representation of connectivity between fragments is advantageous compared to the "Breaking Bridge Bonds" decomposition (Jin et al., 2020; Maziarz et al., 2022), as the separation of motifs, such as rings and chains, does not require truncating the chain. Given this fragmentation, atoms can simultaneously be part of a ring and a chain, and MAGNet accounts for that.

Fig. 6 a illustrate that many fragments share the same topology but differ in the atom and bond types. Extracting a shape $S_i$ from its motif $M_i$ means to reduce the typed adjacency to its binary connectivity, discarding any node features. After creating a vocabulary using all unique shapes from the dataset, we can check their isomorphism by comparing their hashes (Leman & Weisfeiler, 1968).

By this abstraction, MAGNet can learn smoother transitions between different shape representations. Fig. 6 b showcases a simple example of this: the shown motifs share very similar sets of atoms and bonds, as well as their underlying structure, but they differ in the exact positions of atoms and bonds. Fragment-based methods would be required to replace the motif token entirely, having to choose its

replacement from a potentially large vocabulary. By disentangling structure from features, we enable MAGNet to learn such transitions smoothly.

## B   DETAILS MAGNET

### B.1   ENCODER

We build the node features that are processed in MAGNet's encoder from different attributes, see Table 2. We include the atom type ('atom_id_dim'), its charge ('atom_charge_dim'), as well as its multiplicity value ('atom_multiplicity_dim'). We proceed accordingly for the shape level and include the shape id ('shape_id_dim'), its multiplicity ('shape_multiplicity_dim'), as well chemical features ('motif_feat_dim') computed through RDKit (Landrum, 2010). Since the latter are not learned during training, the features are mapped to the specified dimensionality by a linear map.

After processing the resulting node features through the graph transformer (Shi et al., 2021) with 'num_layers_enc'-many layers, they are aggregated in different ways and mapped to specified dimensions as defined by 'enc_<>_dim' for the atoms, shapes, join nodes, and leaf nodes, respectively. On top, the shape embeddings are additionally processed with the same transformer architecture ('num_layers_shape_enc') to inform the embedding about the shape-level connectivity. We then concatenate the resulting graph-level embeddings and further combine them with global molecule features, again computed via RDKit and then mapped to the required dimension ('enc_global_dim'), before mapping them to the latent space via the latent module which has 'num_layers_latent'-many layers.

### B.2   DECODER

All decoding steps are conditioned on the latent code $z$. From $z$, MAGNet employs two transformer decoder layers to autoregressively decode the set of shapes $\mathcal{S}$ by selecting tokens $S_i$ from the extracted shape vocabulary. Generation of the variable-sized multiset $\mathcal{S}$ ends with selecting a stop token. In the next step, an MLP predicts the connectivity $A$ matrix between individual nodes in a permutation-invariant manner. This prediction is solely based on the learnable shape token and multiplicity embeddings. Indicating multiplicity is required, as multiple shapes in $\mathcal{S}$ share the same shape type but have to be connected in different ways.

At this point, MAGNet instantiates the atom-level graph by expanding shape tokens to their untyped graph objects. These graphs without features are then first assigned atom types by transformer decoder layers. Subsequently, the atom features and the respective shape embeddings are used by an MLP to assign bond types independently for every edge, thus creating $\mathcal{M}$. For any connection $A_{ij}$ between two motifs $M_i$ and $M_j$, another MLP then determines the join matrix $J^{(k,l)}$, that is used to identify the shared join node which is then "collapsed". This process applies only to atoms of the same type, is subject to valency constraints, and has to adhere to the predicted join node type $A_{ij}$.

After constructing the core molecule $\mathcal{C}$, MAGNet creates meaningful node embeddings by employing a graph neural network on the core molecule. These embeddings are the basis for a final module consisting of transformer decoder layers that equips the core molecule's atoms with leaf nodes $\mathcal{L}$. By the definition of the leaf nodes, every core molecule's atom can only have one leaf node. A leaf node prediction includes the node's atom type and the bond type connecting it to its attachment atom in the core molecule. The predicted bond type is again subject to valency constraints.

### B.3   ANALYSIS OF ACTIVE UNITS IN THE LATENT SPACE

Formally, the VAE optimises the ELBO

$$ L = \mathbb{E}_{z \sim \mathbb{Q}} \big[ \mathbb{P}(\mathcal{G} \mid z) \big] + \beta D_{\text{KL}} \big( \mathbb{Q}(z \mid \mathcal{G}) \mid P \big) \qquad \text{with} \quad P \sim \mathcal{N}(0, \mathbb{1}) $$

where the posterior $\mathbb{Q}(z \mid \mathcal{G})$ is regularised towards the Normal prior $P$. In practice, finding a balance between the reconstruction loss $\mathbb{E}_{z \sim \mathbb{Q}} \big[ \mathbb{P}(\mathcal{G} \mid z) \big]$ and the KL-divergence $D_{\text{KL}}$ is challenging. Yeung

et al. (2017) and Burda et al. (2015) observe that optimising this objective can result in the VAE learning to collapse several units to the prior to compensate for few non-Gaussian components that support reconstruction. Behaviour like this can be measured through the number of active units in the latent space, defined as $\text{Cov}_{\mathcal{G}}\left(\mathbb{E}_{z \sim \mathbb{Q}(z|\mathcal{G})}[z]\right) > 0.02$ (Burda et al., 2015).

Due to generating the entire molecular context at each generation step, MAGNet heavily relies on the latent representation; also, our reconstruction experiments § 4.1 support this. However, this intended behaviour requires the approximate posterior $\mathbb{Q}(z \mid \mathcal{G})$ to be close to the Normal prior $P$ to allow for good-quality samples. Although there are several methods available to improve the alignment between the approximate posterior and the prior, such as latent dropout (Yeung et al., 2017), a cyclic $\beta$-annealing schedule (Fu et al., 2019), and the GECO loss (Rezende & Viola, 2018), none of them have been able to achieve a rate of active units over 50 % beyond a simple weighting of the $D_{\text{KL}}$ term. As a result of this analysis, we fitted a normalising flow to the VAE, which was trained with low KL regularisation. For this, we follow the framework of Conditional Flow Matching (Lipman et al., 2023; Tong et al., 2023) and achieve 100 % active units.

## B.4 MAGNET: HYPERPARAMETERS AND TRAINING

Training MAGNet for one epoch takes around 30 minutes on a single 'NVIDIA GeForce GTX 1080 Ti'. We trained MAGNet for 30 epochs and fitted the latent normalizing flow post-hoc for 5000 epochs in total and conducted a random hyperparameter sweep including the learning rate, beta annealing scheme, and the number of layers for the encoder and latent module. The MAGNet model reported in the main text has 12.6 M parameters and its configuration is depicted in Table 2. In its current version, MAGNet processes roughly 70 molecules per second during training and samples about 8 molecules per second during inference.

Table 2: Parameter configuration of the best MAGNet runs.

| | Parameter | Value | | Parameter | Value |
|---|---|---|---|---|---|
| **Train** | batch_size | 64 | **Model** | node_aggregation | sum |
| | flow_batch_size | 1024 | | num_layers_latent | 2 |
| | lr | $3.07 \times 10^{-4}$ | | num_layers_enc | 2 |
| | lr_sch_decay | 0.9801 | | num_layers_shape_enc | 4 |
| | flow_lr | $1 \times 10^{-3}$ | | num_layers_hgraph | 3 |
| | flow_lr_sch_decay | 0.99 | | | |
| | flow_patience | 13 | loss_weights | joins | 1 |
| | gradclip | 3 | | leaves | 1 |
| **Model** dim_config | latent_dim | 100 | | motifs | 1 |
| | enc_atom_dim | 25 | | hypergraph | 1 |
| | enc_shapes_dim | 25 | | | |
| | enc_joins_dim | 25 | beta_annealing | max | 0.01 |
| | enc_leaves_dim | 25 | | init | 0 |
| | enc_global_dim | 25 | | step | 0.0005 |
| | atom_id_dim | 25 | | every | 2500 |
| | atom_charge_dim | 10 | | start | 2000 |
| | atom_multiplicity_dim | 10 | | | |
| | shape_id_dim | 35 | | | |
| | shape_multiplicity_dim | 10 | | | |
| | motif_feat_dim | 50 | | | |
| | shape_hidden | 256 | | | |
| | shape_gnn_dim | 128 | | | |
| | motif_seq_pos_dim | 15 | | | |
| | leaf_hidden | 256 | | | |
| | latent_flow_hidden | 512 | | | |

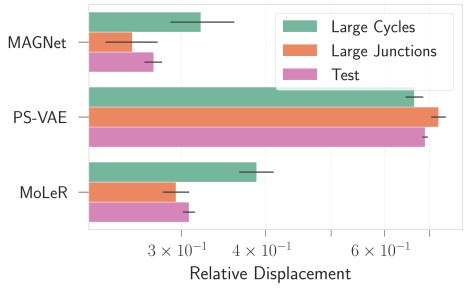 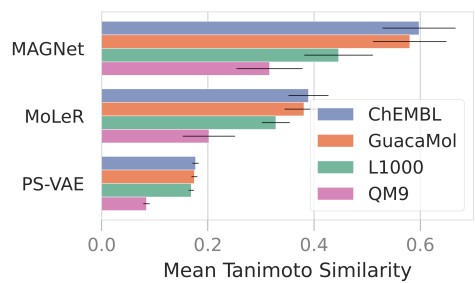

(a) Displacement between latent representation of the input vs. the decoded output.

(b) Tanimoto similarity for zero-shot reconstruction on unseen datasets.

Figure 7: Additional quantitative evaluation shows that MAGNet is faithful to its latent code. From this follows decoding consistency even in challenging cases and on unseen datasets.

## C  ADDITIONAL EXPERIMENTS

### C.1  DISPLACEMENT OF LATENT CODES

To quantify the discrepancy between input and reconstructed molecule visible in Fig. 3a, we measure the displacement of latent codes in Fig. 7a. That is, we obtain the latent representation for the input molecule, decode this latent representation into the output molecule and then obtain the latent representation for the output molecule. This verifies what can be observed qualitatively in Fig. 3a–the evaluated baselines can not reliably decode complex shapes.

### C.2  INTERPOLATION

Extending on Fig. 5, we additionally provide examples for latent space interpolation in Fig. 8. During interpolation, MAGNet stays faithful to the shapes present in the input molecules. The last row shows a failure case of MAGNet: it identifies a shape multiset that can not be fully connected to a molecule.

### C.3  TRANSFERABILITY OF SHAPES

We calculate the Tanimoto similarity in the reconstruction setting for a variety of datasets, Fig. 7b. For all evaluated datasets, MAGNet achieves the best similarity scores between molecules, highlighting the transferability of shapes across various distributions.

We compute the Tanimoto scores only for those molecules that can be represented via the shapes that were extracted from the ZINC dataset. For the QM9 dataset, MAGNet can represent roughly 75% of the molecules in the dataset. This is due to unseen shapes which make up around 11% out of the total number of 289,966 shapes. For GuacaMol, MAGNet can represent around 97% of the molecules in the dataset. Out of the 9,562,028 shapes in GuacaMol, only 0.5% are missing from the shape vocabulary extracted from the ZINC dataset. We consider a fragmentation into shapes that is more flexible and translates even better across datasets important future work.

### C.4  MAGNET ABLATION STUDIES

We show additional results for ablations of different parts of the MAGNet model in Table 3, performing the same analysis as done in § 4.2. MAGNet without a normalising flow achieves an FCD score of 0.65, leading to a performance decrease of more than 14%. A similar decrease can be observed for MAGNet with only a binary shape adjacency $A$. This result further verifies that the atom type of a join node $j$ is an important conditioning for the shape allocation $\mathcal{M}$.

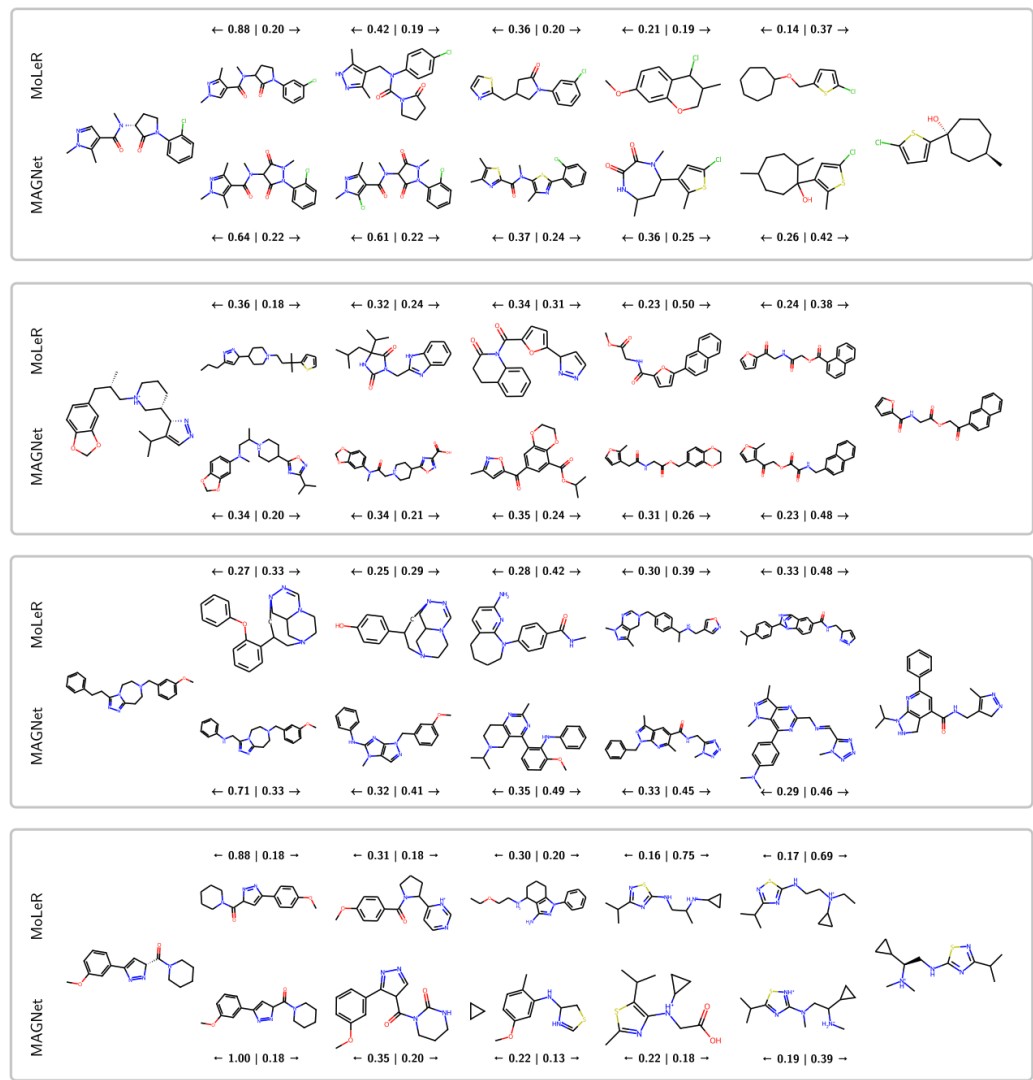

Figure 8: We provide four interpolation examples for MAGNet and MoLeR. The input molecules (left and right) are shared between the two models. We report the Tanimoto similarity as a rough estimate for the interpolation's goodness.

Table 3: GuacaMol and MOSES Benchmark for ablations of MAGNet

|  | FCD ($\uparrow$) | KL ($\uparrow$) | IntDiv ($\uparrow$) | logP ($\downarrow$) | SA ($\downarrow$) | QED ($\downarrow$) |
|---|---|---|---|---|---|---|
| MAGNet | $0.76 \pm 0.00$ | $0.95 \pm 0.00$ | $0.88 \pm 0.00$ | $0.22 \pm 0.01$ | $0.12 \pm 0.01$ | $0.01 \pm 0.00$ |
| no NF | $0.65 \pm 0.00$ | $0.92 \pm 0.00$ | $0.88 \pm 0.00$ | $0.38 \pm 0.06$ | $0.24 \pm 0.03$ | $0.01 \pm 0.00$ |
| Binary $A$ | $0.66 \pm 0.00$ | $0.92 \pm 0.00$ | $0.89 \pm 0.00$ | $0.43 \pm 0.05$ | $0.28 \pm 0.02$ | $0.04 \pm 0.00$ |

