# MAGNet: Motif-Agnostic Generation of Molecules from Shapes

## Abstract

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

Our work addresses this issue by abstracting fragments to shapes, representing only the binary adjacencies, and using this abstraction for vocabulary creation. Since the same shape can lead to multiple distinct atom and bond representations, this effectively reduces the required vocabulary size to capture a dataset's molecules, while still encoding any hard-to-learn structural elements. Simultaneously, we propose an effective generation procedure in the form of our MAGNet model, which learns to generate molecules from abstract shape sets. The shape abstraction enables us sample the *entire* molecule context at each stage and construct the molecular graph in a hierarchical fashion. This hierarchy progresses from shapes and their connectivity to the atom level, where shape representations are generated, ultimately leading to the full definition of the molecule. Notably, our model is the first to freely learn the distribution over shape representations, enabling it to sample a greater variety of atom and bond attributes compared to fragment-based approaches.

While we identify limitations in some of the benchmarks for *effectively* evaluating whether complex structures have been captured, we provide alternative analyses to offer a more comprehensive understanding of the generative behaviour of our and other approaches. Overall, our results highlight MAGNet's ability to reliably reconstruct uncommon shapes, including large cycles, surpassing other methods that face challenges in this area. Additionally, MAGNet excels in sampling a wider range of shape representations, closely aligning with the original data distribution. We also demonstrate MAGNet's advantageous properties, including its fidelity to the presented latent code—a departure from existing methods. Finally, our model enables simultaneous conditioning on multiple scaffolds, underscoring its versatility for diverse applications in molecular generation.

## 2 MODELLING MOLECULES FROM SHAPES

We will first detail the factorisation of the data distribution of molecular graphs, denoted as $\mathbb{P}(\mathcal{G})$. Following this, we will introduce our novel approach to fragmenting molecules into shapes and present the corresponding MAGNet model.

**Factorising the data distribution** $\mathbb{P}(\mathcal{G})$    A molecular graph $\mathcal{G}$ is defined by its structure together with its node and edge features, describing atoms and bonds, respectively. In this work, we consider a factorisation of $\mathbb{P}(\mathcal{G})$ that builds the molecular graph $\mathcal{G}$ from its shape representation, i.e.

$$\mathbb{P}(\mathcal{G}) = \mathbb{P}(\mathcal{G} \mid \mathcal{G}_\mathcal{S}) \, \mathbb{P}(\mathcal{G}_\mathcal{S}) \,,$$

where $\mathcal{G}$ refers to the normal molecular graph and $\mathcal{G}_\mathcal{S}$ to its coarser shape graph. To factorise $\mathbb{P}(\mathcal{G}_\mathcal{S})$ further, we consider a fragmentation into a multiset of shapes $\mathcal{S}$ and their typed connectivity $A \in \{0, \mathrm{C}, \mathrm{N}, \dots\}^{|\mathcal{S}| \times |\mathcal{S}|}$. Each shape $S \in \mathcal{S}$ can be classified as one of three categories, namely (i) rings, (ii) junctions, and (iii) chains. A shape $S$ only holds *structural information* about its $s = |S|$ nodes, meaning we can represent it as a binary matrix, i.e. $S \in \{0, 1\}^{s \times s}$. The shape connectivity $A$, on the other hand, is typed and encodes whether two shapes share an atom, signified by $A \neq 0$, and also the respective atom type of the join, e.g. C or N. Note that $A$ is a shape-level representations which does not hold any information about the exact position of the connection between shapes.

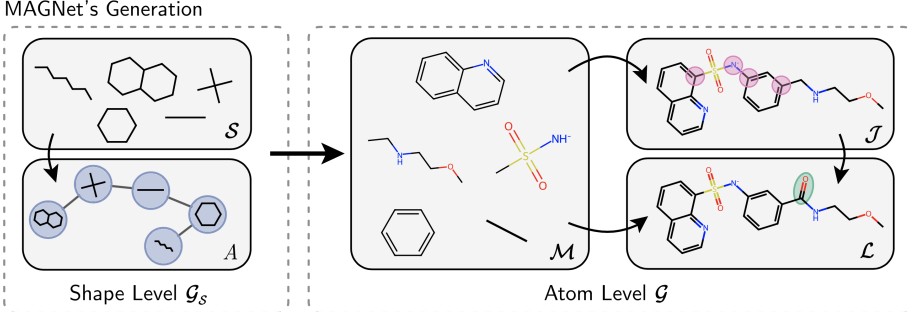

Figure 2: On the shape-level, MAGNet predicts the shape multiset $\mathcal{S}$ and its connectivity $A$. Progressing to the atom-level, $\mathcal{G}_{\mathcal{S}}$ informs the generation of shape representations $\mathcal{M}$. To fully define the molecular graph $\mathcal{G}$, the joining positions $\mathcal{J}$ and the leaf atoms $\mathcal{L}$ are predicted.

Further, since all cyclic structures within the molecular graph are considered individual shapes, $A$ always describes a tree on the shape-level.

Moving forward to the atom-level representation, a shape $S$ can be equipped with node and edges features, corresponding to atom and bond types, respectively. We consider this representation of $S$ to constitute a typed subgraph $M$, or fragment, of the input graph $\mathcal{G}$, and the associated multiset within the molecule $\mathcal{M}$. We note that a shape $S$ can map to multiple distinct $M$ as only the binary adjacencies will be shared between them; we will later on make use of context knowledge to select a suitable $M$.

The shape representations $\mathcal{M}$ and the shape connectivity $A$ do not fully describe $\mathcal{G}$ yet as they do not specify the exact atomic positions where the shapes have to be joined. For this, we define the join set $\mathcal{J}$ which can be understood as the set of join nodes $j$ that are contained in two motifs: $\mathcal{J} = \{j \mid j \in M_k, \ j \in M_l, \ A_{kl} \neq 0, \ S_k, S_l \in \mathcal{S}\}$. Finally, the full molecule is defined by attaching leaf atoms $\mathcal{L}$, which are those atoms with degree $d_i = 1$ and with neighbour $j \in \mathcal{N}_i$ with $d_j = 3$.

In conclusion, $\mathbb{P}(\mathcal{G})$ is factorised into the shape representations $\mathcal{M}$, the join set $\mathcal{J}$, and the leaf set $\mathcal{L}$:

$$\begin{aligned}
\mathbb{P}(\mathcal{G}) &= \mathbb{P}(\mathcal{L}, \mathcal{J}, \mathcal{M}) \\
&= \mathbb{P}(\mathcal{L} \mid \mathcal{M}, \mathcal{J}) \mathbb{P}(\mathcal{J} \mid \mathcal{M}, A) \, \mathbb{P}(\mathcal{M} \mid \mathcal{G}_{\mathcal{S}}) \, \mathbb{P}(\mathcal{G}_{\mathcal{S}}) \,,
\end{aligned}$$

where we use $\mathcal{G}_{\mathcal{S}}$ to factorise the distribution. Note that $\mathcal{J}$ is *conditionally independent* of $\mathcal{S}$ given the motifs $\mathcal{M}$. The same applies to $\mathcal{L}$ being conditionally independent of $A$ given $\mathcal{J}$.

During learning, we start with the coarser shape-level, that is $\mathcal{S}$ and $A$, and then proceed to the atom level for $\mathcal{M}$ and $\mathcal{J}$. Finally, we learn the distribution of leaves $\mathcal{L}$. Note that unlike sequential generation methods, this factorisation considers the entire molecule context at all times with only the level of detail increasing.

## 2.1 FRAGMENTATION INTO SHAPES

MAGNet's fragmentation scheme aims to break down a given molecule into clear structural elements. Initially, we remove all leaf atoms $\mathcal{L}$ across the graph, following the approach outlined in previous works (Jin et al., 2020; Maziarz et al., 2022). Leaves are those nodes that have degree $d_i = 1$ and whose neighbouring node $j \in \mathcal{N}_i$ fulfills $d_{\mathcal{N}_i} = 3$. This step helps to divide the molecule $\mathcal{G}$ into cyclic and acyclic parts. Importantly, instead of modelling the connection between two fragments $M_i$ and $M_j$ with a connecting bond, we represent it by a shared atom, referred to as the join atom $\nu \in \mathcal{J}$. One benefit of this approach is that fragments are not truncated and, for example, two cycles which are connected by only one join atom are identified as two cycles instead of a single cycle that is connected to a chain at both ends.

Moreover, in order to reduce the number of required shapes as much as possible, we further decompose the resulting acyclic fragments. To this end, we introduce "junctions" which are defined by a center node with degree three or four present in an acyclic structure. The junction then contains the center nodes as well as its neighbours. This additional refinement reduces the number of fragments by a

factor of three, allowing to classify them into rings, junctions, and chains. On the ZINC dataset (Irwin et al., 2020), consisting of 249,456 compounds, our fragmentation results in 7371 typed subgraphs. When compared to the decomposition BBB procedure (Jin et al., 2020; Maziarz et al., 2022), our approach reduces complexity by collapsing acyclic structures into distinct shapes, leading to a reduction in vocabulary size by more than a half. Additionally, in contrast to data-driven methods like those outlined in Kong et al. (2022) and Geng et al. (2022), our decomposition method maintains structural integrity through a top-down approach.

Abstracting these typed subgraphs further to shapes ultimately results in 347 distinct shapes, where—in the best case—up to almost 800 fragments are consolidated into a single shape token. While the generative model is required to map a single shape to all its various representations, this fragmentation also enables us to model smoother transitions between shape representations. For instance, consider the molecules "C1NNCC1" and "C1NCNC1", which only differ in the relative positioning among atoms. Using shapes, the model can account for this difference without selecting potentially dissimilar tokens from a large vocabulary. We discuss the potential of our approach for transferability across datasets in Section 4.4 and Appendix C.

## 2.2 MAGNET'S GENERATION PROCESS

MAGNet is designed to represent the hierarchy of the factorisation into shape- and atom-level from Section 2, cf. Fig. 2. The model is trained as a VAE model (Kingma et al., 2015), where the latent vectors $z$ are trained to encode meaningful semantics about the input data. That is, given a vector $z$ from the latent space, MAGNet's generation process first works on the shape level to predict $\mathcal{G}_S$, defined by the multiset $S$ its connectivity $A$, before going to the atom level which is defined by the fragments $\mathcal{M}$, joins $\mathcal{J}$, and leaves $\mathcal{L}$.

**Shape-Level** On the shape-level, MAGNet first generates the **shape multiset** $S$—the same shape can occur multiple times in one molecule—from the latent representation $z$. More specifically, we learn $\mathbb{P}(S \mid z)$ by conditioning the generation on the latent code $z$ and the intermediate representation of the shapes by a transformer model. On the sorted shape set, the network is optimised via Cross-Entropy (CE) loss, denoted by $L_S$.

Given the shape multiset $S$, MAGNet infers the **shape connectivity** $A$ between shapes $S_i, S_j \in S$. Formally, we learn $\mathbb{P}(A \mid S, z) = \prod_{i,j=1}^{n} \mathbb{P}(A_{ij} = t \mid S, z)$ where $t \in \{0, C, N, \dots\}$ not only encodes the existence (or absence) of a shape connection but also its atom type. Further, we assume the individual connections, $A_{ij}$ and $A_{lk}$, to be independent given the shape multiset $S$ and the latent representation $z$. MAGNet implements this connectivity module using an MLP, optimised with CE loss $L_A$. We compute the loss on the shape level as $L_{\mathcal{G}_S} = L_S + L_A$. It is worth noting that the same shape can have different atom representations depending on its positions in the molecular graph, highlighting the importance of predicting $A$ for generating atom-level shape representations.

**Atom-Level** Leveraging the shape set $S$ and connectivity $A$, which together define a molecule's shape-level representation, MAGNet transitions to the atom-level by discerning appropriate node and edge attributes for each shape, referred to as **atom and bond types** $\mathcal{M}$. This step is pivotal in MAGNet's generation process, granting it independence from a pre-established fragment set and thus enhancing its flexibility. To model the shape representation $\mathbb{P}(M_i \mid S, A, z)$ of shape $S_i$, the atoms are generated using a transformer model that constructs its memory by encoding the shape graph and incorporating embeddings tailored to each individual shape (Shi et al., 2021). Unlike the shape set, the sizes of the shapes are fixed, enabling a structured representation of each position within the shape.

Subsequently, the resulting atom embeddings are leveraged to determine the corresponding **bond types** $M^b$ between connected nodes. This bond determination is achieved through the utilisation of an MLP, which effectively captures the necessary relationships for assigning correct bond types. Note that conditioning on $A$ ensures that $M_k$ includes all atoms required for connectivity also on the atom-level, i.e. the atom allocations for $M$ have to respect all join types defined by the shape connectivity $A$: $A_{kl} \in \bigcup_j M_k^a \cap M_l^a$, where $a$ signifies exclusive consideration of atoms. We optimise both atoms and bonds with a CE loss and denote the combined loss by $L_{\mathcal{M}}$.

Next, to establish connectivity on the atom level within the molecular graph, MAGNet proceeds to identify the **join nodes** $\mathcal{J}$. The join nodes' types are already determined by the connectivity $A$. MAGNet accomplishes this by predicting the specific atom positions $p_a$ that need to be combined, collectively forming the join set $\mathcal{J}$. The likelihood of merging nodes $i$ and $j$ in shape representations $M_k$ and $M_l$ is represented by the merge probability $J_{ij}^{(k,l)} = \mathbb{P}(p_i \equiv p_j \mid \mathcal{M}, A, z)$, which constitutes the join matrix $J^{(k,l)} \in [0,1]^{V_{S_k} \times V_{S_l}}$. It is modelled and optimised using an MLP and CE loss $L_{\mathcal{J}}$.

Finally, predicting the **leaves** $\mathcal{L}$ involves two key aspects: determining the correct atom type for each leaf and establishing its connection to the molecule's atom representation, denoted as $\mathcal{C}$ for core molecule. To learn $\mathbb{P}(\mathcal{L}_S \mid \mathcal{C}, z)$, we assess each node position within the shape representation $M_S$. Similar to the approach employed in modelling the shape representation $P(M_i \mid S, A, z)$ for shape $S_i$, MAGNet utilises a transformer model to generate atom representations, optimised with CE loss $L_{\mathcal{L}}$. This model constructs its memory by encoding the present atom graph and incorporating embeddings tailored to each unique shape representation. The final atom loss is defined by $L_{\mathcal{G}} = L_{\mathcal{M}} + L_{\mathcal{J}} + L_{\mathcal{L}}$.

## 2.3 THE MAGNET ENCODER

MAGNet's encoder aims to learn the approximate posterior $\mathbb{Q}(z \mid \mathcal{G})$. At its core, the encoder leverages a graph transformer Shi et al. (2021) for generating node embeddings of the molecular graph. Since MAGNet generates molecules in a coarse to fine-grained fashion, it is beneficial to encode information about the decomposition of the molecules. To achieve this, we use an additional GNN to capture the coarse connectivity within the shape graph. The embeddings of the molecular and the shape graph are computed by aggregating over the individual atom and shape nodes, respectively. In addition to these aggregations, we exclusively aggregate joins and leaves to represent the other essential components of the graph. The representation of the individual components—molecular graph, shapes, joins, and leaves—are concatenated and mapped to the latent space by an MLP, constituting the graph embedding $z_{\mathcal{G}}$. More details about the chosen node features as well as technical specifications of the encoder can be found in Appendix D.

Taken together, we optimise MAGNet according to the VAE setting, maximising the ELBO:

$$L = \mathbb{E}_{z \sim \mathbb{Q}}\big[\mathbb{P}\big(\mathcal{G} \mid z\big)\big] + \beta D_{\mathrm{KL}}\big(\mathbb{Q}(z \mid \mathcal{G}) \mid P\big) \qquad \text{with} \quad P \sim \mathcal{N}(0, \mathbb{1})$$
$$= L_{\mathcal{G}_S} + L_{\mathcal{G}} + \beta D_{\mathrm{KL}}\,,$$

where the KL-divergence $D_{\mathrm{KL}}$ serves to regularise the posterior $\mathbb{Q}(z \mid \mathcal{G})$ towards similarity with the Normal prior $P$ in the latent space, weighted by $\beta$. However, we have observed that this regularisation alone is inadequate for achieving a smoothly structured latent space. To address this, we apply a Normalizing Flow to the latent space, aligning it more effectively with the prior (Tong et al., 2023).

## 3 RELATED WORK

**Molecule generation** Existing generative models can be broadly divided into two categories: (1) string-based models, relying on string representations like SMILES or SELFIES (Adilov, 2021; Fang et al., 2023; Flam-Shepherd et al., 2022; Grisoni, 2023; Gómez-Bombarelli et al., 2018; Segler et al., 2018), which do not leverage structural information, and (2) graph-based models, which are inherently centered around molecular graphs. Graph-based approaches involve models that represent molecular graphs (1) primarily at the atom level and (2) predominantly through fragments. Regarding the generation process, they can be further divided into sequential methods (Ahn et al., 2021; Assouel et al., 2018; Bengio et al., 2021; Jin et al., 2020; Kajino, 2019; Khemchandani et al., 2020; Li et al., 2018; Lim et al., 2020; Liu et al., 2018; Luo et al., 2021; Mercado et al., 2021; Popova et al., 2019; Shi et al., 2020; Shirzad et al., 2022; Yang et al., 2021; You et al., 2019), building molecules per fragment while conditioning on a partial molecule, and all-at-once (AAO) approaches (Bresson & Laurent, 2019; De Cao & Kipf, 2018; Flam-Shepherd et al., 2020; Kong et al., 2022; Liu et al., 2021; Ma et al., 2018; Samanta et al., 2019; Simonovsky & Komodakis, 2018; Zang & Wang, 2020) that create each aspect of the molecular graph in a single step.