# OpenReview forum: "MAGNet: Motif-Agnostic Generation of Molecules from Shapes"
_ICLR.cc/2024/Conference — Submitted to ICLR 2024_

### Official Review · Reviewer_bmZE · 2023-10-27

**Soundness:** 2 fair
**Presentation:** 1 poor
**Contribution:** 2 fair
**Rating:** 3
**Confidence:** 3

**Summary:**

The paper proposes a fragment-based VAE molecule generative model MAGNet to capture the full spectrum of molecules. Different from previous fragment-based models which use fixed fragment vocabulary and struggle to generate uncommon structures, MAGNet decomposes the generation process into two levels, i.e., shape-level and atom-level, for more flexible fragment decisions. Specifically, on the shape-level, MAGNet first generates the coarse shape/topology of molecules. Then, on the atom-level, MAGNet fills up the shape with atoms to get actual molecules. Experiments demonstrate that MAGNet can generate more topologically distinct structures and outperforms most graph-based approaches on standard benchmarks.

**Strengths:**

- The paper is well-motivated. While fragments are different in their atom constitution, they may share the same shape/topology with each other. Adding a shape-level generation step makes us select fragments more flexibly and capture the full spectrum of molecules more easily.
- The paper is relatively novel. MAGNet’s two-level generation style provides a reasonable way of designing molecules which as far as I know is different from main stream molecule generation methods.
- Authors provide extensive experiment results showing MAGNet can capture the full spectrum of molecules.

**Weaknesses:**

- Not including a related work section discussing the connection between MAGNet and existing shape-based molecular generation methods [1, 2, 3].
- The proposed method is not very clear to me. See details in Questions.

[1] Long, S., Zhou, Y., Dai, X., & Zhou, H. (2022). Zero-shot 3d drug design by sketching and generating. Advances in Neural Information Processing Systems, 35, 23894-23907.

[2] Adams, K., & Coley, C. W. (2022). Equivariant shape-conditioned generation of 3d molecules for ligand-based drug design. arXiv preprint arXiv:2210.04893.

[3] Chen, Z., Peng, B., Parthasarathy, S., & Ning, X. (2023). Shape-conditioned 3D Molecule Generation via Equivariant Diffusion Models. arXiv preprint arXiv:2308.11890.

**Questions:**

- I do not understand how MAGNet generates the shape multiset mentioned in section 2.2. Can you give me more details?
- Also in section 2.2, when inferring the shape connectivity, does MAGNet need to provide atom type? Why does the atom type exist on the shape-level step?
- How does MAGNet extract fragments/shapes? Can you provide more examples in section 2.1?
- In section 2.3, how you apply a Normalizing Flow to the latent space is confusing.
- The paper uses the term shape to describe the intermediate results which I personally think the term topology is more appropriate.

---

> ### Author Response · Authors · 2023-11-17
> **Response to Reviewer bmZE**
>
> &nbsp;
>
> Dear Reviewer,
>
> We greatly appreciate your time and effort in reviewing our manuscript. Your feedback has been valuable in improving our work, and we address your concerns in our answer below. Your questions regarding clarifications of the proposed generation approach aided in revising the methods section. In our answers, we indicate where we changed the manuscript to address the raised concerns.
>
> &nbsp;
>
> > Connection between our work and other "shape" works that were not discussed
>
> - Note that the mentioned works focus on a different task where shapes refer to surfaces in 3D space while we consider untyped 2D graph as shapes. We are familiar with these works but originally refrained from including them, to avoid confusion with respect to the task we address. Adams et al., in particular, use MoLeR to create sufficient priors for their method. The same is possible with our approach and, due to the introduction of shapes, even with more flexibility.
> We realise that not including these works also has the potential for confusion. Hence, we differentiate our methods from the mentioned "shape" works in the updated version of the related work section, cf. end of Section 3.
>
> &nbsp;
>
> In the following, we discuss your questions and concerns regarding the method description.
>
> > How do you generate the shape multiset?
>
> - We train a 2-layer transformer decoder to generate the shape multiset. We use the decoder to learn the shape set in an autoregressive manner. We updated the "shape-level" paragraph in Section 2.2 and provide a more detailed explanation of the multiset generation in the Appendix B.2.
>
> > Why is an atom type present on the shape level?
>
> - The requirement of the atom type on the shape's adjacency helps to ensure that the generated shape allocations can be joined on the atom level. Strictly speaking, this is not required and we updated the "Factorising the data distribution" paragraph in Section 2 accordingly. We analysed the effect of this design choice and provide the results for this experiment here. Including the atom type of the join improves the quality of generated molecules by 16% according to the FCD score. We explain the reason for this design choice more thoroughly in the "shape-level" paragraph of Section 2.2.
>
> > How do we extract the shapes? Can you provide more examples?
>
> - We clarified the discussion about the fragmentation and extraction of shapes in Section 2.1 of the manuscript. We further provide additional samples of the shape set in Appendix A, where we illustrate examples that we only discussed verbally in the initial manuscript.
>
> > How the Normalising flow is applied is confusing
>
> - We observed that solely optimising the KL-term leads to over-pruning [1]. Thus, we apply a normalising flow to the latent space. To do so, we rely on Conditional Flow Matching [2, 3].  We have clarified this point at the end of Section 2.3 of the revised manuscript.
>
> > Terminology: shapes -> topology
>
> - We apologise for the confusion caused by the terminology. After internal discussion, we decided to stick with the shape term in the current version of the manuscript. However, we provided a clarifying statement at the beginning of Section 2, "Factorising the data distribution", when defining shapes and refer to the concept of topology in various other parts other the discussion.
>
> &nbsp;
>
> We hope to have addressed your concerns and answered your questions adequately and we are looking forward to discussing our work further with you.
>
> &nbsp;
>
> [1] Serena Yeung, Anitha Kannan, Yann Dauphin, Li Fei-Fei, Tackling Over-pruning in Variational Autoencoders, arxiv.org/abs/1706.03643
> [2] Yaron Lipman, Ricky T. Q. Chen, Heli Ben-Hamu, Maximilian Nickel, and Matthew Le, Flow matching for generative modeling, International Conference on Learning Representations, 2023, arxiv.org/abs/2210.02747
> [3] Alexander Tong, Nikolay Malkin, Guillaume Huguet, Yanlei Zhang, Jarrid Rector-Brooks, Kilian Fatras, Guy Wolf, and Yoshua Bengio, Improving and generalizing flow-based generative models with minibatch optimal transport, arxiv.org/abs/2302.00482

---

> > ### Comment · Reviewer_bmZE · 2023-11-21
> >
> > Thank you for your comprehensive responses. While most of my concerns have been addressed, I still find the paper challenging to grasp due to the scattered presentation of details. An improved overview section could be beneficial in this regard. Additionally, the paper should include ablation studies to demonstrate the contribution of each MAGNet component, e.g., normalising flow, atom type on the shape level. I can only recommend acceptance of the paper after a significant revision. At this stage, I will maintain my current score.

---

> ### Author Response · Authors · 2023-11-21
> **Follow-Up to Reviewer bmZE**
>
> Dear Reviewer,
>
> Thank you for your response. We appreciate your feedback. However, it remains unclear to us what requires significant revision. Below, we provide detailed answers to the raised points to clarify our understanding of the feedback.
>
> &nbsp;
>
> > I still find the paper challenging to grasp due to the scattered presentation of details.
>
> Based on the feedback we received, we believe that our (revised) manuscript follows a clear structure. Hence, we are unsure about the scattered details that were mentioned in the feedback. If the criticism is related to referring to the appendix multiple times from the main manuscript, we would like to clarify that the appendix A only provides additional examples and supporting information that are not crucial for the main part of the manuscript and were asked for in the initial review. The same applies, for example, to the appendix sections B which provides implementation details that will become less relevant once the code is publicly available. We hope this clears up any confusion.
>
> > An improved overview section could be beneficial in this regard.
>
> We would like to understand why the provided overview paragraphs are insufficient. We include multiple overview paragraphs throughout the main manuscript to help guide the reader:
> - In the introduction, we clearly state our contributions, followed by a detailed explanation of the claims made in the paper and how they are reflected in the experiment section.
> - Section 2 introduces our contributions on the methodological level by first discussing the data factorisation and nomenclature, followed by our fragmentation procedure and MAGNet as a generative model fitting the fragmentation and proposed factorisation. This structure is again highlighted at the beginning of Section 2.
> - In Section 3, after discussing key aspects of our manuscript in two paragraphs of related work, we put MAGNet in a broader context and distinguish our work from others which also was refined based on the feedback we got in the the initial review.
> - Section 4, the experiment section, starts again with an overview of the different experiments, introducing their aspects and outlining how they are meant to analyse key aspects of the generative approach.
>
> > Additionally, the paper should include ablation studies to demonstrate the contribution of each MAGNet component, e.g., normalising flow, atom type on the shape level.
>
> We are happy to include ablation studies as we already have provided the results for the atom type inclusion in our intial response. Regarding the normalising flow, we find the analysis on active units and insufficiency of the KL-regularisation more convincing. However, we have also analysed the generative performance of a MAGNet model without the normalising flow:
> - We observed that the performance for the experiments from section 4.1 and 4.3 and 4.4 remained largely unchanged.
> - The unconditional generation, however, as analysed in 4.2 decreased to a score of 0.65 according to the FCD metric.
>
> As these analyses do not directly reinforce the primary arguments of the paper, we have included them in appendix section C.4.
>
> &nbsp;
>
> In our understanding, the raised criticism pertains solely to the details of the method and stylistic aspects of the manuscript, not its content or main contributions. We believe that the current score does not reflect this adequately, and we would be grateful if you would reconsider your evaluation.

---

> ### Comment · Reviewer_bmZE · 2023-11-22
>
> In essence, the paper's comprehensiveness is hindered by two key factors:
>
> - **Unfamiliar Terms**: Several technical terms, such as 'leaves,' 'joins,' and 'OS,' are introduced without adequate explanation. These terms appear in Section 2.2 and Table 1, respectively, leaving readers without a clear understanding of their usage and significance.
> - **Model Design Rationale**: The paper lacks a thorough explanation of the rationale behind the model's design. The current presentation resembles a summary of experiments rather than a comprehensive discussion of the model's architecture and underlying principles. For instance, the justification for adopting a two-stage approach for shape generation instead of a more end-to-end approach remains unclear.
>
> While I acknowledge the authors' efforts and the potential contributions of their work, the current presentation of the paper falls short of the bar of ICLR. A revised overview section that clearly defines key terms and elaborates on the design principles of the model would significantly enhance its comprehensibility. In the absence of such revisions, I maintain my current score.

---

> > ### Author Response · Authors · 2023-11-22
> > **Follow-Up to Reviewer bmZE**
> >
> > Dear Reviewer,
> >
> > Thank you for the prompt response and the clarification, as well as engaging in the discussion with us.
> >
> > &nbsp;
> > > Several technical terms, such as 'leaves,' 'joins,' and 'OS,' are introduced without adequate explanation
> >
> > We would like to point out that we have defined all mathematical objects in the introduction of Section 2, "Factorising the data distribution". However, we realise that we used similar terms, such as leaves/leaf nodes/leaf atoms, interchangeably, which might have been confusing. We have updated the manuscript with consistent terminology, particularly "leaf nodes" and "join nodes" throughout the text for better consistency. To further guide the reader, we have provided a pointer to the related work section, where we discuss the terminology "OS / sequential" of Table 1.
> >
> > > A revised overview section that clearly defines key terms and elaborates on the design principles of the model would significantly enhance its comprehensibility.
> >
> > Thank you for the clarifying comment, which has prompted us to restructure the introduction of our manuscript. We have paid attention to clarify the structure of our work better and highlight the rationale that motivates our work. Connected to this, we have also rewritten the beginning of Section 2, "Factorising the data distribution", to better guide the reader through the definition of the mathematical objects which are used in the subsequent sections (Section 2.1 ", Identifying a concise set of shapes from the data" and Section 2.2, "MAGNet's generation process").
> >
> > &nbsp;
> >
> > To facilitate your review of these changes, we have highlighted them in orange. We hope to have addressed all remaining concerns to your satisfaction.

---

### Official Review · Reviewer_qQTC · 2023-10-29

**Soundness:** 3 good
**Presentation:** 3 good
**Contribution:** 3 good
**Rating:** 8
**Confidence:** 4

**Summary:**

The paper introduces MAGNet, a novel graph-based model for molecule generation, which uniquely generates abstract shapes before allocating atom and bond types. This approach is proposed to increase flexibility across datasets and address the limitations of existing models that rely heavily on motif representations. The authors provide a comprehensive explanation of the model, its generation process, and the underlying methodology, demonstrating its improved performance on standard benchmarks and its ability to generate diverse molecular structures.

**Strengths:**

Innovation: The paper presents a novel approach to molecule generation, moving away from the traditional motif-based methods. The introduction of abstract shapes as an intermediate step in the generation process is a significant innovation, potentially leading to more flexible and diverse molecule generation.

Comprehensive Methodology: The authors provide a detailed explanation of the MAGNet model, its generation process, and the underlying methodology. The factorisation of the data distribution of molecular graphs and the hierarchical generation process from shapes to atom and bond types are well-articulated.

Improved Performance: The paper demonstrates that despite the added complexity of shape abstractions, MAGNet outperforms most other graph-based approaches on standard benchmarks. This is a strong point in favor of the proposed method.

Diversity in Molecule Generation: The authors highlight and demonstrate that MAGNet’s improved expressivity leads to the generation of molecules with more topologically distinct structures and diverse atom and bond assignments, which is crucial for applications in drug discovery and material science.

**Weaknesses:**

1. Insufficient Comparative Analysis:
The paper falls short in providing a comparative analysis with diffusion-based models, which are crucial in the domain of molecule generation. This lack of comparison might lead to an incomplete evaluation of MAGNet, as readers are left without a clear understanding of how the proposed model stands against these advanced alternatives. A thorough comparison, highlighting the strengths, weaknesses, and performance differences, would significantly enhance the paper’s credibility and provide a more comprehensive assessment of MAGNet.

2. Deviation from Established Terminology:
The terminology used to describe MAGNet’s generation process, specifically the terms "sequential" and "Attribute-Atom-Object (AAO)," deviates from the established "Sequential" and "One-shot" terminology commonly used in the field. This inconsistency could potentially create confusion and hinder the paper’s accessibility to readers familiar with the standard terms. Adopting the widely accepted terminology would ensure clarity and maintain consistency across the literature.

3. Limited Reference to Foundational Works:
The paper does not adequately reference foundational works and comprehensive surveys in the field of molecule generation, which could limit the readers’ ability to place MAGNet within the broader context of the field. Including references to seminal works such as "A systematic survey on deep generative models for graph generation," and "A survey on deep graph generation: Methods and applications," would provide a richer background, enhancing the paper’s credibility and informative value.

4. Exclusion of Alternative Molecule Representations:
The exclusive focus on a graph-based representation for molecules in the paper neglects the discussion of other popular representations like fingerprints. This exclusion limits the comprehensiveness of the paper, as readers are not provided with a comparison or rationale for the chosen representation. Discussing various molecule representations, their advantages, disadvantages, and applicability, would contribute to a more holistic view of the field and strengthen the paper’s content.

**Questions:**

Would like to improve the authors have address the weakness above.

---

> ### Author Response · Authors · 2023-11-17
> **Response to Reviewer qQTC**
>
> &nbsp;
>
>  Dear Reviewer,
>
> Thank you for taking the time to review our work and providing us with valuable feedback. We appreciate your constructive criticism and insightful comments that will help us improve our manuscript.
>
> We are pleased to observe that you appreciate the strengths of our work. Regarding the weaknesses you have pointed out, we have noted them and addressed them point-by-point. Additionally, we indicate where we changed the manuscript to address the respective concern.
>
> &nbsp;
>
> > Comparison with diffusion models
>
>
> - We initially decided against adding a paragraph on diffusion models as these primarily focus on molecule generation in 3D space [1,3,4], which is different from the focus of our work. However, we agree that it is beneficial for the reader to present the entire spectrum of molecule representations (SMILES, graphs, geometries) and have added in a paragraph in Section 3 (Related Work, "Molecule generation") of the updated manuscript.
> - We highlight DiGress [2] specifically as it is a discrete diffusion model for general graphs that is applicable to 2D molecular graphs. However, similar to SMILES-LSTM, a comparison between the models is difficult due to its implicit learning of representations. DiGress, in its current form, is not applicable to the majority of our experimental evaluations and downstream applications, such reconstruction experiments in Section 4.1 and 4.3, scaffold/shape conditioning in Section 4.4, or the interpolation between molecules in Appendix C.
> - However, DiGress works for unconditional generation and we will include it in the presented benchmark, see Section 4.2. After training for >2,5 days on one A100 GPUs (already 2x the compute resources of any other model), the model achieves an FCD score of 0.6 (MAGNet has 0.76). For the benchmark computation, we discarded all invalid molecules, which corresponds to about 26% of the generated samples. Note that sampling is 33x slower than MAGNet.
>
> > Missing surveys citations
>
> - We are aware of these works [5,6,7] and apologise for the oversight. To put MAGNet in a broader context, we updated the manuscript accordingly, see Section 3 "Related work" and Section 1 "Introduction".
>
> > Unusual terminology
>
> - We acknowledge your comment regarding terminology and updated our manuscript to follow the survey from Zhu et al. [6], that is, we replace "AAO" with "OS". Importantly, "AAO" refered to "all-at-once" (Section 3) rather than "all-atom-object", which was motivated by the work of Yang et al. [5].
>
>
> > Discuss the variety of representations for molecules
>
> - In the initial manuscript, we focused on the differentiation between SMILES and graphs for molecule generation. We added a clarifying sentence on using descriptors for molecule representation in Section 3 to address your comment. For this, we followed the survey from Du et al. [7].
>
> &nbsp;
>
> We appreciate your feedback and hope we addressed all raised concerns to your satisfaction in the revised manuscript. Once again, thank you for your time and effort in reviewing our work.
>
> &nbsp;
>
> [1] Emiel Hoogeboom, Victor Garcia Satorras, Clément Vignac, Max Welling, Equivariant Diffusion for Molecule Generation in 3D, arxiv.org/abs/2203.17003
> [2] Clement Vignac, Igor Krawczuk, Antoine Siraudin, Bohan Wang, Volkan Cevher, Pascal Frossard, DiGress: Discrete Denoising diffusion for graph generation, arxiv.org/abs/2209.14734
> [3] Clement Vignac, Nagham Osman, Laura Toni, Pascal Frossard, MiDi: Mixed Graph and 3D Denoising Diffusion for Molecule Generation, arxiv.org/abs/2302.09048
> [4] Minkai Xu, Alexander Powers, Ron Dror, Stefano Ermon, Jure Leskovec, Geometric Latent Diffusion Models for 3D Molecule Generation, arxiv.org/abs/2305.01140
> [5] Nianzu Yang, Huaijin Wu, Junchi Yan, Xiaoyong Pan, Ye Yuan, Le Song, Molecule Generation for Drug Design: a Graph Learning Perspective, arxiv.org/abs/2202.09212
> [6] Yanqiao Zhu, Yuanqi Du, Yinkai Wang, Yichen Xu, Jieyu Zhang, Qiang Liu, Shu Wu, A Survey on Deep Graph Generation: Methods and Applications,arxiv.org/abs/2203.06714
> [7] Yuanqi Du, Tianfan Fu, Jimeng Sun, Shengchao Liu, MolGenSurvey: A Systematic Survey in Machine Learning Models for Molecule Design, arxiv.org/abs/2203.14500

---

> > ### Comment · Reviewer_qQTC · 2023-11-17
> >
> > Thanks for the response. All my concerns have been addressed.

---

> ### Author Response · Authors · 2023-11-21
> **Follow-Up to Reviewer qQTC**
>
> Dear Reviewer,
>
> Thank you for the kind response and the update of your score. For completeness, we would like to follow up on the intermediate results that we have reported for the DiGress model. At the time of reporting, the model had been training for 2.5 days and achieved an FCD score of 0.6. After an additional 2 days and completing 1000 epochs of training, this has not further improved.
>
> Again, we would like to thank you for your valuable feedback.

---

### Official Review · Reviewer_dbRP · 2023-10-30

**Soundness:** 3 good
**Presentation:** 3 good
**Contribution:** 3 good
**Rating:** 6
**Confidence:** 4

**Summary:**

The authors present MAGNet, a graph-based generative model with novel molecular structure factorization. Unlike other fragment-based representation learning methods, the authors utilize abstract fragment shape templates. Shape-level structure abstract shows superior performance over fragment-level representations, which is reasonable.

In general, I find this manuscript clearly-written and easy to follow. The idea of shape-based factorization regularizes the vocabulary space to a limited number of shape templates, thus removing a large number of "chemical degeneracy" (i.e., different combinations of atomic species over the same structure template). It is reasonable that this method helps neural networks effectively learn the shape distribution and generate diverse molecular structures. I suggest accepting this manuscript after addressing some minor concerns discussed below.

**Strengths:**

- Factorization of molecules into abstract (untyped) shapes is a reasonable and smart idea. Chemistry-agnostic shapes provide higher-level abstraction than conventional motifs, and help the representation learning model perceive molecular shape distributions.
- Intuitively, MAGNet effectively decouples modeling structural and chemical diversity into two consecutive generation steps, hence the corresponding neural network only needs to capture distributions in a subspace instead of their product space. This may explain the reported diversity and validity of generated molecules.
- Similar strategies have been adopted in protein design, e.g., RFDiffusion, where one first generates the backbone structure, then design a sequence that can fold into this structure.
- Fig. 6 in Appendix is interesting. The smooth transition/interpolation between two distinct molecules shows the advantage of using shape templates over motif templates.

**Weaknesses:**

See questions.

**Questions:**

- I am curious about the reason behind not being able to effectively train the VAE model (i.e., prior matching term), but have to fit another NF to the latent space. Would you please provide more analysis here?
- My understanding is that the model has been trained in an end-to-end manner. Atom-level inference is conditioned on the same latent vector $z$ as used in shape-level generation. If the latent space is also decoupled, would it help improve diversity of generated molecules as well as obtain a stable prior matching loss?
- Fig 3b, does the x-axis show percentage (e.g., 0.50%) or ratio? Also, fragment-based methods model the distribution of shape&atomic species. Therefore, it is reasonable that if you marginalize over the atomic species and only analyze the shape coverage, MAGNet will by design do better at diverse shape sampling.
- Could you please provide more details on fitting normalizing flow for the latent space?
- Consider a particular generation task where a specific motif is desired to be present in the generated molecule, can MAGNet be tailored for this task?

---

> ### Author Response · Authors · 2023-11-17
> **Response to Reviewer dbRP**
>
> &nbsp;
>
> Dear Reviewer,
>
> We appreciate your comprehensive review of our manuscript and valuable feedback. It is encouraging to see that you appreciate our work in multiple ways. We answer your questions below and further indicate how we addressed your feedback in the revised manuscript.
>
> &nbsp;
>
> > Unclear why KL-term does not work and how normalising flow is fitted
>
> - It is important to note that we conducted an extensive investigation into the ineffectiveness of the KL-term. During our analysis of the latent space, we found that optimising the KL-term leads to over-pruning, meaning many latent factors fail to learn anything and become inactive [1]. Despite trying different methods, such as introducing dropouts or epitomes, we could not solve this optimisation issue satisfactorily. As a result of this analysis, we chose to fit a normalising flow post-hoc to the VAE which was trained with low KL regularsation. We agree that this clarification would be beneficial to include in the manuscript and have updated Section 2.3 accordingly, providing the analysis of active units in Appendix B.3 as well as an ablation study on this in Appendix C.4.
> - Regarding details on the normalising flow, we clarify its optimisation in the revised version of Section 2.3 and put it in context with the above explanations.
>
> > Idea: decouple shapes and atom-level also in latent space
>
> - We appreciate your suggestion regarding decoupling shapes and atom-level in latent space. As the computed features with respect to shape and atom graph are extracted from the same molecule, they will be correlated. Therefore, it is not easily possible to disentangle the representations for this hierarchy. However, we followed up on your suggestions and split the latent dimensions to decode for the distinct properties of the molecular graph according to our factoriation. Here, we investigated the setting of splitting into shape and atom representations. In our experiment, the performance of our MAGNet model was unchanged, and we hypothesise that modelling shape and atom level truly independently, one has to change the datastructure (train the hierarchy differently) and/or introduce explicit disentangling losses.
>
> > Fig 3b
>
> - Thank you for your feedback on Fig 3b's axis label. The figure had a typo, and we intended to refer to the ratio. As you correctly pointed out, MAGNet is designed to capture the structural variety of shapes. However, note that this design choice introduces additional complexity for generating shape allocations, which MAGNet captures to such an extent that it even outperforms other methods, highlighting its greater flexibility shown in Figure 4. We demonstrate MAGNet's ability to learn meaningful atom allocations both qualitatively (4a) and quantitatively (4b). To clarify this point, we updated Section 4.1 ("MAGNet reliably decodes shapes") of our manuscript.
>
> > Condition MAGNet on a particular motif
>
> - One can readily use MAGNet for this kind of conditional task. We have shown results for this task in Figure 5 (top) of the initial manuscript. As this might have been unclear before, we improved the experiment discussion in the corresponding paragraph of Section 4.4.
> &nbsp;
>
> Once again, we thank you for your insightful feedback and hope we have addressed your concerns satisfactorily in the revised manuscript.
>
> &nbsp;
>
> [1] Serena Yeung, Anitha Kannan, Yann Dauphin, Li Fei-Fei, Tackling Over-pruning in Variational Autoencoders, arxiv.org/abs/1706.03643

---

> > ### Author Response · Authors · 2023-11-21
> > **Follow-Up to Reviewer dbRP**
> >
> > Dear Reviewer,
> >
> > Thank you again for your valuable feedback, which was essential to improving the quality of our work.
> >
> > We hope that our response has adequately addressed your concerns. If so, we would appreciate it if you could raise your score. If this is not the case, we would appreciate further feedback and will continue actively responding to your concerns.

---

> > > ### Comment · Reviewer_dbRP · 2023-11-21
> > > **Response to author rebuttal**
> > >
> > > Dear authors,
> > >
> > > Thanks for your response, which addressed most of my concerns. I went through other reviews and decide to maintain my score due to the following reasons: [1] The idea to use abstract shape representation is novel and reasonable; [2] Yet I agree with questions raised by reviewer jTCd regarding its applicability to downstream drug discovery tasks (beyond improving benchmark performance).

---

### Official Review · Reviewer_jTCd · 2023-10-30

**Soundness:** 2 fair
**Presentation:** 1 poor
**Contribution:** 1 poor
**Rating:** 1
**Confidence:** 4

**Summary:**

This work presents MAGNet, a generative model of small molecule graphs that uses a hierarchy of internal representations of subgraphs that make up the molecule.  The model first uses a simple connectivity pattern of nodes and edges, then names the atoms and specific bond types, and constructs the full molecule.  The authors demonstrate the use of MAGNet on existing datasets and discuss some properties of the generated molecule sets.

**Strengths:**

This paper has an original design of a pre-training model that makes some sense conceptually. The demonstration of conditional design prompted by a fragment or a shape is interesting as are the interpolation examples in the appendix. Unfortunately, the bulk of the paper does not come close to what I'd expect from an ICLR publication. It is hard to make a case for the significance of this work; the benchmarks are rather weak and to a degree uninteresting for practical drug discovery as there are no downstream applications of importance (logP design is trivial; SA/QED is mildly OK, but these are also simple to learn) and no uses of large pretraining sets. Given that this is ICLR I'd have liked to see something more detailed about the internal representations of this model, but there wasn't much to read about them. It also hurts the paper that the presentation of the method is unnecessarily unclear (the language and notation at the bottom of page 2 is messy but just defines the Murcko scaffolds, which one can explain concisely; more broadly much of section 2 could be denser/simpler.)

**Weaknesses:**

If I understand correctly, then what the authors call "shape" is the so-called Murcko scaffold of individual fragments. Murcko scaffold basically is: turn every atom into carbon and every bond into a single one. It would have helped if the authors clarified this point early on.

The listed benchmark snippets in table 1 are not super relevant to drug discovery and the comparison models are weak compared to what one could nowadays train on a single GPU during a single week on a large dataset.  It would be nice to see the model applied to the open graph benchmark.

There is no good explanation of why one would like to use this model instead of simpler generalizable architectures, ideally pretrained on huge datasets.  I can understand the problems with generalization of the early JTVAE models, but the present model also inherently has such limitations, even if they are dramatically less likely (say if would not make a sterol if no structure with four fused rings existed in its training set.)

**Questions:**

Why didn't the authors train this model on a large dataset (think 10s or 100s of millions of molecules) to make sure they can compare edge cases against other models, if any?

Do the authors have a reasonable argument for why at scale it would help to have their model instead of say a simple transformer model?

The hierarchical design of the model is somewhat interesting, especially given the known problems of vanilla GNNs to spontaneously discover these hierarchies themselves.  Did the authors take a view at the internal layers of their model to see examples of how the model represents molecules internally, and perhaps gain insight for how to scale up this model?

Did the authors try to apply their model to the open graph benchmark (both the large-scale benchmark and the property prediction components that use small molecules)?  At least in that way it would be easier to demonstrate promise of the model architecture.

---

> ### Author Response · Authors · 2023-11-17
> **Response to Reviewer jTCd**
>
> &nbsp;
>
> Dear Reviewer,
>
> We have acknowledged your feedback and are surprised by the tone of the review. Nevertheless, we believe many of the points raised in this review are based on misunderstandings, which we would like to clarify.
>
> &nbsp;
>
> > Questions on the General Relevance of the GuacaMol/MOSES benchmarks for Drug Discovery, Evaluation on the Open Graph Benchmark
> - Our experimental section focuses on critically assessing how well the underlying distribution is learned. We employ two standard benchmarks for this purpose. The GuacaMol distribution learning benchmark "encompasses measuring the fidelity of the models to reproduce the (property) distribution of the training sets" [1]. Further, the MOSES benchmark evaluates "the quality and diversity of generated structures" [2]. Both benchmarks are specifically designed for de-novo molecular generation models. These benchmarks have become standard for 2D de-novo molecule generation over the past few years, with most works published at ICLR or similar venues relying on them for critical evaluation of their methodology [3,4,5,6].
> - MAGNet challenges a common perception in the domain of graph-based molecule generation: the requirements of fragments. We discuss the limitations of the standard benchmarks in Section 4.2 and complement the benchmark results comprehensively in the rest our experimental section.
>
> > Why did you not train on large-scale benchmarks, do the authors have a reasonable argument for why at scale it would help to have their model
> - The "Open Graph Benchmark" is a benchmark for property prediction on the edge, node, and graph level, it is not a benchmark for graph generation. The encoder we use as part of MAGNet has already been evaluated on the OGB benchmark [7].
>  - The core motivation for the abstraction to shapes is that representing a distribution of molecules by motifs suffers from increasing combinatorial complexity (by also considering edge and node features). As we show in our experimental evaluation, the ZINC dataset is sufficient to expose drawbacks to fragment-based methods and demonstrate the advantages of our factorisation as demonstrated by MAGNet.
>  - We are certain that these factors are applicable in the case of a large amount of data as well. For instance, the MAGNet model only requires a set of less than 400 shapes to decode for more than 98% of the 1.5 million molecules in the GuacaMol dataset, whereas the MICAM model requires more than 20,000 motifs for the ZINC dataset alone.
>  - Our work provides a new perspective on fragment-based generation methods. While we are interested in further improving our method, e.g. the shape extraction process towards an even more concise vocabulary, our experimental evaluation has shown the superior performance of MAGNet. We consider the task of "scaling up 2D molecule generation methods" a natural step for future work which is, however, beyond the scope of the current manuscript.
>
> > Look at the internal representation
> - We would be keen to know in what way such an analysis would add additional insights beyond the already conducted experiments.
>
> > What the authors call "shape" is the so-called Murcko scaffold of individual fragments
> - MAGNet does not operate with Murcko scaffolds. Rather than utilising the simplest feature allocation (Murcko scaffolds), MAGNet works with featureless abstract graph objects (shapes). We deemed this more suitable for the audience of an ML conference like ICLR. Nevertheless, this misunderstanding about MAGNet's use of shapes has prompted us to improve Section 2 in a general fashion.
>
> &nbsp;
>
> We appreciate your feedback and hope to have addressed all raised concerns in the revised manuscript. We look forward to your response.
>
> &nbsp;
>
> [1] Brown et al., GuacaMol: Benchmarking Models for de Novo Molecular Design, Journal of Chemical Information and Modeling, 2019
> [2] Polykovskiy et al., Molecular sets (MOSES): a benchmarking platform for molecular generation models, Frontiers in pharmacology, 2020
> [3] Kong et al., Molecule Generation by Principal Subgraph Mining and Assembling, Advances in Neural Information Processing Systems, 2022
> [4] Geng et al., De Novo Molecular Generation via Connection-aware Motif Mining, International Conference on Learning Representations, 2023
> [5] Maziarz et al., Learning to Extend Molecular Scaffolds with Structural Motifs, International Conference on Learning Representations, 2022
> [6] Jin et al., Hierarchical Generation of Molecular Graphs using Structural Motifs, International Conference on Machine Learning, 2020
> [7] Shi et al., Masked label prediction: Unified message passing model for semi-supervised classification, 2020

---

> > ### Comment · Reviewer_jTCd · 2023-11-22
> > **Thanks for the additional effort.**
> >
> > Thank you for the additional clarifications.  I understand that GuacaMol and MOSES are consistently used in a large fraction of the small-molecule graph generation literature, but I question their use in evaluating the power of a model for practical drug discovery applications. The types of rings that show up in many of the generative models by such molecules are not appearing in typical/easy medicinal chemistry, and the questions of what are good pretraining datasets and what are better evaluation methods for selecting practical tools are open.
> >
> > I was hoping that if your model learns a useful internal representation one could then finetune it into a property predictor to at least show an advance in public tests like the Open Graph Benchmark.  To me, such a demonstration would at least show that I'd need to pay more attention to the underlying concept and maybe give it the benefit of the doubt despite the lack of demonstration of generatlization outside the domain of pre-trained fragments/shapes and the possibility that the model would simply not work if it encountered totally new molecules in its expanded future set.  I understand that even the most powerful current neural network do not naturally abstract away the graphs into its own most-meaningful subgraphs and we now suspect that existing GNN are not up to the task, so a method along the lines you're working on could eventually be useful, but nothing in the paper convinced me about this possible future development.  Having a look at how the internal representations handle sets of related molecules might show insights that would make it interesting again.
> >
> > Regarding fragmentation more generally, I wouldn't say that fragmentation is a requirement for molecular generation; although it seems reasonable as an idea, there certainly exist excellent molecule generators out there that don't need to fragment molecules (they only need a large training set).  I reread your modified section of the paper describing the shapes and stand by my previous understanding that what the current method does it to use Murcko scaffolds of fragments as inputs.  Here is your new text: "For this, we start by removing all leaf atoms L across the graph,..." and earlier "These fragments are typed, and MAGNet abstracts them further into shapes by removing atom and bond types, resulting in 347 distinct shapes."  This is exactly the definition of the Murcko scaffold for a fragment.  Again, it's not so important a point, but it would have helped.  The new test is somewhat improved, but still hard to parse correctly.
> >
> > I still think that the method will break when it encounters a different new macrocycle, or a fused multicycle construct.  Maybe it is a little more robust than the junction-tree VAE for typical/common molecules, but its failure modes will be similar, if perhaps less common.
> >
> > Overall the contribution does not meet the bar of ICLR in my mind, however, it seems that other reviewers were more open to considering this work for possible inclusion.  I've updated my previous score to reflect the somewhat improved language in the new draft, but I still cannot accept this work at this conference.

---

### Author Response · Authors · 2023-11-17
**General Response to Reviewers**

&nbsp;

We are pleased to have received positive feedback on our work, particularly noting its motivation, novelty, and superior performance in the extensive evaluation.

We appreciate the reviewers' constructive feedback and insightful questions, which have prompted us to clarify the manuscript further.

&nbsp;

The changes in the pdf are indicated by the purple colour, with added citations having a slightly different shade. They include:
- Revising our method section for clarity, mostly streamlining it and connecting it better with the introduction.
- Providing additional clarification on why MAGNet benefits from a normalising flow fitted to the latent space, outlining the methodology. Refer also to our answer to reviewer dbRP for details.
- Restructuring the Appendix for better coherence, including a more detailed explanation of the fragmentation and abstraction to shapes. This is supported by additional examples and implementation details.
- Incorporating additional related work on 3D molecule generation, diffusion models, and surface-guided molecule generation.

&nbsp;

We are looking forward to a constructive discussion with the reviewers.

---

### Meta-Review · Area_Chair_RVuL · 2023-12-08

**Metareview:**

**Summary**
This paper proposes a graph generative model primarily designed for molecular generation. In graph-based molecular generation, it is a common practice to use motifs (subgraphs) of molecules as building blocks and connect them to construct novel molecules. To enable more flexible generation, this paper takes a step further by decomposing such motifs into shapes. These shapes comprise non-labeled graphs representing graph topological structures and features to retain information about atom and bond types. The proposed method, called MAGNet, is a VAE based model that uses this hierarchical approach from shapes to molecules to generate new molecules.

**Strengths**
- The proposed shape based approach is reasonable. The whole pipeline of the proposal is carefully designed.
- The proposal has been thoroughly evaluated on experiments. It is good that both of two popular benchmarks, GuacaMol and MOSES, are included.

**Weaknesses**
- This paper's presentation, especially regarding mathematical notations, is problematic, which significantly deteriorates the overall quality of the paper. For example, graphs are never properly defined. Random variables, graphs, vertices, and other concepts are mixed up and not properly used.
- Due to the above lack of clarity, understanding the proposed methodology of the paper is challenging, as highlighted by the reviewers. This indicates that not only is clarity an issue, but the overall quality and significance of the paper are also not sufficiently convincing.

**Justification For Why Not Higher Score:**

The weaknesses of the paper, as outlined above, are crucial and must be addressed for the publication of this paper. As I believe that the technical contribution of this paper is potentially interesting, I strongly advise addressing all the raised issues by the reviewers for substantial improvement before resubmission.

**Justification For Why Not Lower Score:**

N/A

---

### Decision · Program_Chairs · 2024-01-16

Reject